# Beyond Neighbors: Distance-Generalized Graphlets for Enhanced Graph Characterization

## Abstract

Graphs are widely used to model complex systems across various domains, including social networks and biological systems. A key task in graph analysis is identifying recurring structural patterns, known as graphlets, which capture connectivity among a fixed-size subset of nodes. While graphlets have been extensively applied in tasks such as measuring graph similarity and identifying communities, conventional graphlets focus only on direct connections between nodes. This limitation overlooks potential insights from more distant relationships within the graph structure.

In this paper, we introduce $(d, s)$-graphlets, a generalization of size-$s$ graphlets that incorporates indirect connections between nodes up to distance $d$. This new formulation provides a more fine-grained and comprehensive understanding of local graph structures. To efficiently count $(d, s)$-graphlets in a graph, we present EDGE, an exact counting algorithm that employs optimized combinatorial techniques to significantly reduce computational complexity compared to naive enumeration. Our empirical analysis across diverse real-world datasets demonstrates that $(d, s)$-graphlets provide superior graph characterization, outperforming conventional graphlets in the graph clustering task. Moreover, our case studies show that $(d, s)$-graphlets uncover non-trivial insights that would remain undiscovered when using conventional graphlets.

### ACM Reference Format:
Anonymous Author(s). 2018. Beyond Neighbors: Distance-Generalized Graphlets for Enhanced Graph Characterization. In *Proceedings of Make sure to enter the correct conference title from your rights confirmation email (Conference acronym 'XX)*. ACM, New York, NY, USA, 17 pages. https://doi.org/XXXXXXX.XXXXXXX

## 1 INTRODUCTION

Graphs are widely used to model complex systems across various domains, from social networks to biological systems. A key task in understanding and predicting the behaviors of these systems is identifying recurring structural patterns, which can provide insights into their underlying dynamics.

Among the various approaches, *graphlets* [49, 50] describe connectivity patterns among a small set of nodes (typically 3, 4, or 5 nodes). Graphlets capture local structures within a graph, and real-world graphs can often be distinguished by their domain, or from random graphs, based on the occurrence patterns of the graphlets.

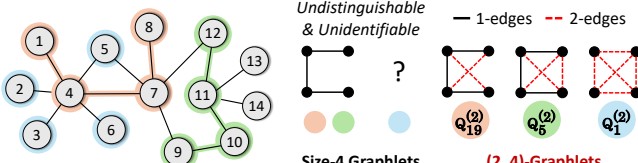

**Figure 1: A sample graph and three sets of 4-node subgraphs. The red dotted lines indicate that the distance between two nodes is 2. If there is no line between two nodes, it means the distance between them is greater than 2. Graphlets, which only account for direct connections between nodes, (1) cannot distinguish between subgraphs such as {1, 4, 7, 8} and {9, 10, 11, 12}, and (2) cannot describe subgraphs with disconnected nodes such as {2, 3, 5, 6}. Our proposed $(d, s)$-graphlets address these limitations by considering relationships that extend beyond direct connections (distance $\geq 2$), allowing for more fine-grained and comprehensive local structure analysis.**

In practice, the occurrences of each graphlet within a given graph are counted [53], and these counts are then used to measure graph similarity [56], detect anomalies [6, 28], classify nodes [18, 33], or identify communities [8, 37, 73].

While graphlets are defined to capture connectivity patterns based only on direct connections in a general context of graph analysis, both traditional and recent studies have highlighted the potential of exploring relationships beyond direct connections. The significance of relationships between nodes that are not directly connected (i.e., at a distance of 2 or larger) has long been recognized in social science to enhance the contextual interpretation of nodes [39]. More recently, incorporating non-neighboring nodes has been shown to offer key benefits across multiple domains, including improved feature representation in machine learning tasks, with applications in biology [4], recommendation systems [17], and general graph machine learning methods [24, 64, 70, 72].

Motivated by these insights, in this paper, we introduce **$(d, s)$-graphlets**, a novel generalization of size-$s$ graphlets that accounts for indirect connections between nodes up to distance $d$ (see Figure 2 in Section 4). We first define $d$-edges, which generalize edges by representing relationships between non-neighboring nodes at a distance of $d$. Using these higher-order connections, we define $(d, s)$-graphlets to describe the local connectivity pattern, incorporating all 1-edges to $d$-edges while distinguishing connections based on their distances. This extension allows for a more fine-grained and comprehensive analysis of local graph structures, revealing patterns that would otherwise remain undiscovered with conventional graphlets. An example is shown in Figure 1, where $(d, s)$-graphlets effectively capture and distinguish local structural patterns, while simple graphlets fail to differentiate or identify them.

Our comprehensive analysis using 13 real-world datasets from 5 different domains reveals that $(d, s)$-graphlets are highly effective at

capturing local structural patterns. Specifically, the relative counts of each $(d, s)$-graphlet, when compared to those of null models, show better differentiation between graphs from different domains compared to simple graphlets. This enhanced characterization highlights the importance of modeling relationships beyond immediate neighbors for a more accurate analysis of local graph structures.

As a means to count the occurrences of each $(d, s)$-graphlet in a graph, we develop **EDGE** (**E**xact Counting of **D**istance-**G**eneralized Graphl**e**ts), an algorithm for exactly counting instances of each $(2, 3)$-, $(3, 3)$-, and $(2, 4)$-graphlets. To avoid exhaustive enumeration, EDGE categorizes $(d, s)$-graphlets into non-deducible, semi-deducible and deducible $(d, s)$-graphlets based on structural properties. It selectively enumerates instances of non/semi-deducible $(d, s)$-graphlets, and using their counts, rapidly computes the count of deducible $(d, s)$-graphlets through combinatorial methods without enumeration. Moreover, EDGE employs a specialized directed acyclic graph that models relationships between nodes up to distance $d$, further enhancing its speed and scalability.

To summarize, our contributions are:

- **New concept.** We introduce a novel definition of graphlets by generalizing them to additionally consider relationships between non-neighboring nodes (Section 4).
- **Algorithm.** We develop an efficient algorithm for exactly counting the occurrences of each $(d, s)$-graphlet. EDGE is up to $14.86\times$ faster than a naive counting method (Section 5).
- **Discoveries.** Using the counts of $(d, s)$-graphlets, we demonstrate that they exhibit strong characterization power in distinguishing real-world graphs (Section 6).

**Reproducibility.** The code and datasets used in this work are *anonymously* available at [26].

## 2 RELATED WORK

In this section, we review previous work relevant to our study.

**Local structural patterns and graphlets.** Mining local structural patterns from graphs is a common approach for understanding the underlying dynamics of complex systems [20, 30, 68]. A key challenge is identifying structural properties that distinguish real-world graphs from random ones, as these distinctions provide valuable insights into the behavior and organization of such systems. Among the various analytic tools for graph analysis, graphlets [49, 50] have been effective in characterizing network structures. As fundamental building blocks of graphs, the counts of graphlets serve as characteristic measures used to assess graph similarity [56], detect anomalies [6, 28], classify nodes [18, 33], and identify communities [8, 37, 73]. Recently, graphlets have also been leveraged to enhance the graph machine learning techniques [15, 19, 33]. Graphlets have been extended in various directions by incorporating node or edge labels [27, 55], node attributes [57], edge weights [63], and multi-layer structures [10, 52]. However, existing definitions focus on direct connections between nodes and overlook the potential insights from examining indirect (i.e., multi-hop) connections.

**Graphlet counting algorithm.** Various methods have been proposed to count graphlets in a graph. Early approaches enumerate all connected subgraphs with a small number of nodes [41, 43, 66, 67]. To improve scalability and avoid exhaustive enumeration, more recent methods take a more analytical approach. These methods

**Table 1: Frequently-used notations.**

| Notation | Definition |
|---|---|
| $G = (V, E)$ | a graph with nodes $V$ and edges $E$ |
| $\delta(u, v)$ | distance between nodes $u$ and $v$ |
| $E^{(d)}$ | the set of $d$-edges (the distance between nodes is $d$) |
| $E^{(\leq d)}$ | the set of $\{1, 2, \cdots, d\}$-edges (i.e., $\{E^{(1)}, \cdots, E^{(d)}\}$) |
| $N_u^{(d)}$ | the set of $d$-neighbors of node $u$ |
| $\vec{N}_u^{(d)}$ | the set of out-going $d$-neighbors of node $u$ |
| $G^{(d)} = (V, E^{(\leq d)})$ | $d$-graph of the graph $G$ |
| $\vec{G}^{(d)} = (V, \vec{E}^{(\leq d)})$ | $d$-DAG (directed acyclic graph) of the graph $G$ |
| $C(g; G)$ (or $C(g)$) | the counts of $(d, s)$-graphlet $g$ in a graph $G$ |

exploit the relationships between the counts of different graphlets and deduce the count of some graphlets based on the counts of others [1, 2, 29, 40, 48]. For example, PGD [1, 2] and ESCAPE [48] decompose graphlets into smaller primitives and use their counts to derive the count of the larger ones using combinatorial arguments. This approach significantly improves the scalability of graphlet counting and the size of the graphlets that can be counted. Specifically, PGD can count up to size-4 graphlets, and ESCAPE extends this capability to count graphlets up to size-5. Moreover, approximate counting methods for graphlets, such as path sampling [65], random walk [71], and color coding [16], make a trade-off by sacrificing some accuracy to gain time efficiency. We take an approach based on exact counting algorithms, not approximate ones, to ensure a precise comparison with conventional graphlets.

**Distance generalization in general graph analysis.** Many prior studies have emphasized the potential of exploring relationships between nodes that are not directly connected by edges [14, 17, 39, 51, 64]. Incorporating relationships between nodes without direct connections (i.e., those at a distance greater than 1) has been shown to enhance the performance on various tasks in domains including natural language processing [9, 22, 51], biology [4], and recommendation systems [17], and improving graph machine learning methods [24, 47, 64, 70, 72]. A natural extension of this idea is to generalize the concept of distance between two nodes to incorporate indirect connections. Conventionally, the degree of a node is defined as the number of its directly connected neighbors. This can be generalized by introducing a distance threshold $d$, where the degree of a node is defined as the number of nodes within a distance of at most $d$. This generalization of degree has led to various extensions of graph mining tools. One of the earliest such generalizations is $d$-clique [5, 39], where every pair of nodes in the clique is within a distance of $d$. Similarly, a $d$-club [5, 44] is defined as a maximal subset of nodes in which the induced subgraph has a diameter of at most $d$. More recently, $k$-cores have been generalized to $(k, d)$-cores [14, 21, 38, 58], which ensure that each node has at least $k$ other nodes within a distance of $d$. These generalizations have uncovered interesting patterns previously undetected, providing deeper insights by extending analysis beyond direct relationships. However, generalizing distances in graphlets has not been explored, which we focus in this paper.

## 3 NOTATIONS & BASIC CONCEPTS

In this section, we discuss the notations and basic concepts that will be used to explain our concepts (Section 4) and algorithms (Section 5). Refer to Table 1 for the frequently-used notations.

**Graphs and distances.** A graph $G = (V, E)$ consists of a set of nodes $V$ and a set of edges $E$. The distance $\delta(u, v)$ between two nodes $u, v \in V$ is defined as the length of the shortest path connecting them. Specifically, if $u$ and $v$ are directly connected by an edge (i.e., $(u, v) \in E$), the distance between them is 1. If no path exists between the two nodes, their distance is considered infinite.

**Induced subgraphs.** Given a set of nodes $S \subseteq V$, the induced subgraph on $S$ is the subgraph $G_S = (S, E_S)$, where $E_S$ is the set of all edges between nodes in $S$ that are present in the original graph $G$, i.e., $E_S = \{(u, v) \in E : u, v \in S\}$.

**Graphlets.** A *graphlet* is an induced subgraph that represents a specific connectivity pattern among a small number of nodes (typically, 3, 4, or 5 nodes). The size of a graphlet refers to the number of nodes it contains. Formally, a graphlet is an equivalence class of such subgraphs under graph isomorphism. Specifically, two induced subgraphs $G_S = (S, E_S)$ and $G_{S'} = (S', E_{S'})$ are considered isomorphic if there exists a bijection $\phi : S \rightarrow S'$ such that for every pair of nodes $(u, v) \in \binom{S}{2}$, the connectivity relationship is preserved, i.e., $(u, v) \in E_S \Leftrightarrow (\phi(u), \phi(v)) \in E_{S'}$. This implies that the connectivity patterns are identical between the subgraphs on $S$ and $S'$ under the mapping $\phi$.

**Null models.** To accurately characterize real-world graphs, we compare them with null models. In this work, we employ random graphs generated by the configuration model [45] as null graphs, which preserves degree distributions of the nodes.

## 4 PROPOSED CONCEPTS

In this section, we propose $(d, s)$-graphlets, which are tools for understanding the local structural characteristics of graphs. We first discuss some specialized concepts and definitions. Using these concepts, we formally define $(d, s)$-graphlets.

### 4.1 Preliminary Concepts

We begin by defining some specialized concepts that are essential for defining $(d, s)$-graphlets in Section 4.2.

**$d$-edges.** We define a *$d$-edge* as a pair of nodes whose distance is $d$ in the graph. Any pair of nodes $(u, v)$ that forms an actual edge in the graph (i.e., $(u, v) \in E$) is referred to as a 1-edge. For node pairs with a distance of $d \geq 2$, they form $d$-edges, where explicit edges do not exist in the original graph, representing *virtual* connections between the nodes. We denote the set of $d$-edges in $G$ by $E^{(d)} := \{(u, v) \in \binom{V}{2} : \delta(u, v) = d\}$. Importantly, the sets $E^{(d)}$ are pairwise disjoint (i.e., $E^{(d)} \cap E^{(d')} = \varnothing$ for all $d \neq d'$) since $\delta(u, v)$ is uniquely defined for each node pair $(u, v)$. We denote the union of all 1-edges through $d$-edges as $E^{(\leq d)} = \{E^{(1)}, E^{(2)}, \cdots, E^{(d)}\}$, and define the *$d$-extended graph* as $G^{(d)} = (V, E^{(\leq d)})$ which includes both actual and virtual connections up to distance $d$. Finally, a node $v$ is called a *$d$-neighbor* of $u$ if $(u, v)$ is a $d$-edge, and we denote the set of $d$-neighbors of $u$ as $N_u^{(d)}$.

**$d$-induced subgraphs.** Given a set of nodes $S \subseteq V$, the *$d$-induced subgraph* on $S$ is the subgraph $G_S^{(d)} = (S, E_S^{(\leq d)})$, where $E_S^{(\leq d)}$ consists of all 1-edges, 2-edges, up to $d$-edges between nodes in $S$ from the original graph $G$. More formally, $E_S^{(\leq d)}$ is defined as: $E_S^{(\leq d)} = \{E_S^{(1)}, E_S^{(2)}, \cdots, E_S^{(d)}\}$, where for each $d' \in \{1, 2, \cdots, d\}$,

the edge set $E_S^{(d')}$ represents the set of all $d'$-edges between nodes in $S$, i.e., $E_S^{(d')} = \{(u, v) \in E^{(d')} : u, v \in S\}$. (Note that the distance between nodes in $G_S^{(d)}$ is measured in the original graph, not in the subgraph $G_S^{(d)}$.) Notably, conventional induced subgraphs are 1-induced subgraphs, which only consider direct connections (i.e., 1-edges) between nodes in $S$. In contrast, $d$-induced subgraphs generalize this concept by capturing higher-order connectivity patterns beyond direct connections.

**$d$-isomorphism.** Given two sets of nodes, $S$ and $S'$, and their $d$-induced subgraphs $G_S^{(d)} = (S, E_S^{(\leq d)})$ and $G_{S'}^{(d)} = (S', E_{S'}^{(\leq d)})$, they are considered $d$-isomorphic if there exists a bijection $\phi : S \rightarrow S'$ such that for every pair of nodes $(u, v) \in \binom{S}{2}$, the following holds:

$$(u, v) \in E_S^{(d')} \Leftrightarrow (\phi(u), \phi(v)) \in E_{S'}^{(d')}, \quad \forall d' \in \{1, 2, \cdots, d\}.$$

This indicates that the mapping $\phi$ preserves the structure of all edges up to distance $d$ between the nodes in $S$ and $S'$, meaning the subgraphs are structurally identical with respect to $d$-edges.

### 4.2 $(d, s)$-Graphlets

We are now ready to present the definition of $(d, s)$-graphlets. We generalize graphlets by incorporating *relationships beyond direct connections* to describe the connectivity patterns of $s$ nodes.

**Definition.** A $(d, s)$-graphlet is a $d$-isomorphism class of size-$s$ $d$-induced subgraphs. More specifically, the $d$-induced subgraphs $G_S^{(d)}$ and $G_{S'}^{(d)}$ of two sets, $S$ and $S'$, each containing $s$ nodes (i.e., $|S| = |S'| = s$), belong to the same $(d, s)$-graphlet if they are $d$-isomorphic. In essence, a $(d, s)$-graphlet represents an equivalence class of $d$-induced subgraphs where the local structure, including both direct and indirect connections up to distance $d$, is identical.

**Examples.** In Figure 2, we present examples of $(d, s)$-graphlets. We let $T^{(d)}$ denote the set of all size-3 $(d, s)$-graphlets (*triplets*) and $Q^{(d)}$ denote that of size-4 $(d, s)$-graphlets (*quartets*). There exist 6 (2, 3)-graphlets ($T_1^{(2)}$ - $T_6^{(2)}$; $|T^{(2)}|$=6), 13 (3, 3)-graphlets ($T_1^{(3)}$ - $T_{13}^{(3)}$; $|T^{(3)}|$=13), and 36 (2, 4)-graphlets ($Q_1^{(2)}$ - $Q_{36}^{(2)}$; $|Q^{(2)}|$=36).

**Comparison with graphlets.** When considering only direct edges (i.e., 1-edges), there are only two types of size-3 graphlets (a triangle and a wedge) and six types of size-4 graphlets (e.g., as a 4-clique or 4-cycle). However, as shown in Figure 2, incorporating edges beyond direct neighbors (e.g., 2-edges and 3-edges) allows for finer distinctions among patterns of 3 or 4 nodes. While increasing the number of nodes in simple graphlets may provide more insights into graph structure, it also exponentially increases the number of graphlet types and thus requiring significantly more complex and computationally expensive counting algorithms. In Section 6, we demonstrate that size-3 and size-4 $(d, s)$-graphlets more effectively characterize graphs compared to larger graphlets.

### 4.3 Characteristic Profiles

To summarize the $(d, s)$-graphlet characteristics of a graph, we use a measure called *characteristic profile* (CP), which is conventionally used in graphlet studies [31, 42, 61, 62, 69]. First, we count the occurrences (i.e., number of instances) of each $(d, s)$-graphlet. Let the occurrence count of the $(d, s)$-graphlet $g$ in graph $G$ be denoted

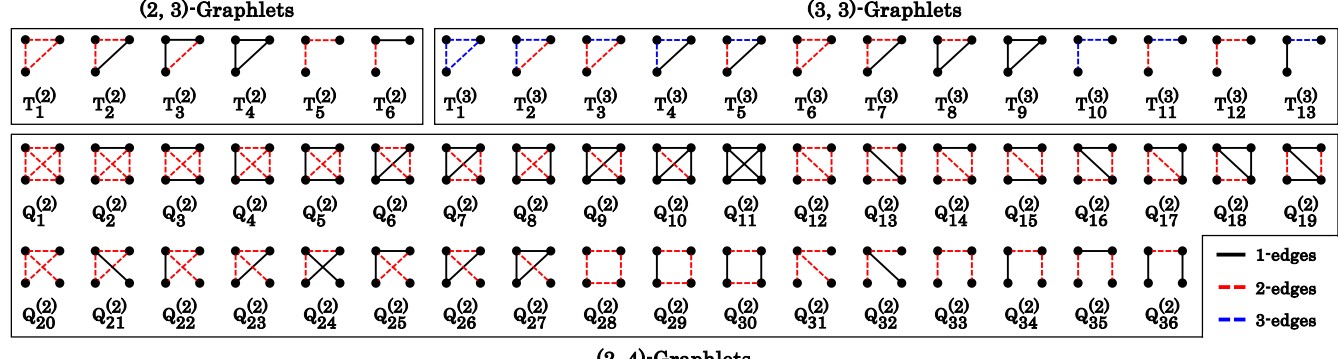

**Figure 2: All the** $(2, 3)$**-graphlets (**$T_1^{(2)}$ **-** $T_6^{(2)}$**),** $(3, 3)$**-graphlets (**$T_1^{(3)}$ **-** $T_{13}^{(3)}$**), and** $(2, 4)$**-graphlets (**$Q_1^{(2)}$ **-** $Q_{36}^{(2)}$**). Solid edges represent actual edges (i.e.,** 1**-edges), while dotted edges represent virtual edges (i.e.,** 2**-edges and** 3**-edges). The** 1**-edges,** 2**-edges, and** 3**-edges are colored in black,** red**, and** blue**, respectively.**

as $C(g; G)$ (or $C(g)$ for brevity). Then, the significance of a $(d, s)$-graphlet $g$ is defined as:

$$\mu_g = \frac{C(g; G) - C(g; G_{\text{rand}})}{C(g; G) + C(g; G_{\text{rand}}) + \epsilon}$$

where $G_{\text{rand}}$ is a randomized graph of $G$ generated by a null model (see Section 3), and $\epsilon$ is a small constant (e.g., $\epsilon = 1$). Based on the significance, the CP of $g$ is computed as the normalized significance:

$$\text{CP}_g = \frac{\mu_g}{\sqrt{\sum_{g' \in \tilde{g}} \mu_{g'}^2}}$$

where $\tilde{g}$ is the set of all considered $(d, s)$-graphlets (e.g., $T^{(2)}$, $T^{(3)}$, or $Q^{(2)}$). The CP is represented as a vector by concatenating the CP values of each $(d, s)$-graphlet, which contains the local structural information of the graph.

## 5 PROPOSED ALGORITHMS

In this section, we present EDGE, our algorithm for the exact counting of $(d, s)$-graphlets in a given graph. While $(d, s)$-graphlets are generally defined for arbitrary values of $d$ and $s$, we focus on three specific configurations: $(d, s) = \{(2, 3), (3, 3), (2, 4)\}$. In Section 6, we empirically demonstrate that these configurations are effective and general enough, compared to simple graphlets with similar sizes, to uncover non-trivial structural patterns within the graph. We first introduce our method for counting size-3 $(d, s)$-graphlets (i.e., $(2, 3)$- and $(3, 3)$-graphlets), followed by our method for counting size-4 $(d, s)$-graphlets (i.e., $(2, 4)$-graphlets).

**Remarks.** The problem of counting $(d, s)$-graphlets (particularly for $d \geq 2$), is computationally more challenging than counting graphlets (i.e., for $d = 1$). Specifically, $(d, s)$-graphlets are defined based on relationships between nodes up to distance $d$, requiring the exploration of $E^{(2)}, \cdots, E^{(d)}$, where the number of edges grows exponentially as $O(\Delta^d)$, where $\Delta$ is the maximum node degree. For example, as shown in Table 4, the number of 3-edges is at most 15× more than that of the original edges (i.e., 1-edges). Moreover, the distance that each edge connects should be accounted for when determining the $(d, s)$-graphlet of an instance. These unique challenges incur significant bottlenecks for exhaustive enumeration

methods, and thus efficient and specialized algorithms for counting $(d, s)$-graphlets without direct enumeration are demanded.

### 5.1 Graph Construction

For efficient $(d, s)$-graphlet counting, EDGE constructs a directed acyclic graph (DAG) that consists of actual edges (connecting immediate neighbors) and virtual edges (connecting nodes beyond their immediate neighbors), as a common preprocessing step.

$d$-**Graph construction.** Firstly, EDGE constructs virtual edges that connect nodes at distances beyond their immediate neighbors. Specifically, it builds additional edge sets $E^{(2)}, \cdots, E^{(d)}$, resulting in a $d$-graph $G^{(d)} = (V, E^{(\leq d)})$. This process is performed using a breadth-first search (BFS) traversal, as outlined in Algorithm 3 (see Appendix A.1). The time complexity of this step is given in Lemma 1, and the proof is provided in Appendix A.1.

**Lemma** 1 (Complexity of $d$-Edge Construction). *The time complexity of constructing $d$-edges for a graph $G = (V, E)$ is $O(|V|\Delta^d)$, where $\Delta$ is the maximum degree of the graph.*

$d$-**DAG construction.** Once the (undirected) $d$-graph $G^{(d)}$ is constructed, EDGE builds a $d$-degree-ordered directed acyclic graph (DAG) of $G$, referred to as a $d$-DAG. Specifically, for each distance $d' \in \{1, \cdots, d\}$, it creates a directed edge $(u, v)$ if $u \prec^{(d)} v$, where $\prec^{(d)}$ represents the degree ordering based on the $d$-edges, implying $|N_u^{(d)}| \leq |N_v^{(d)}|$. The resulting $d$-DAG is denoted as $\vec{G} = (V, \vec{E}^{(\leq d)})$, where $\vec{E}^{(\leq d)} = \{\vec{E}^{(1)}, \cdots, \vec{E}^{(d)}\}$, and $\vec{E}^{(d')} = \{(u, v) \in E^{(d')} : u \prec^{(d)} v\}$ for each $d' \in \{1, \cdots, d\}$. For a node $u$, $\vec{N}_u^{(d)}$ denotes the out-going neighbors of $u$ at distance $d$, i.e., $\vec{N}_u^{(d)} = \{v : (u, v) \in \vec{E}^{(d)}\}$. Importantly, the number of out-going neighbors is typically smaller than the number of neighbors in undirected graphs (i.e., $|\vec{N}_u^{(d)}| \ll |N_u^{(d)}|$), which significantly contributes to improving the scalability of EDGE, as empirically demonstrated in Section 6. For more details, refer to Appendix A.2.

### 5.2 Size-3 $(d, s)$-Graphlet Counting

We now explain how EDGE counts size-3 $(d, s)$-graphlets (i.e., $s = 3$), specifically focusing on $(2, 3)$-graphlets ($T^{(2)}$) and $(3, 3)$-graphlets

---

**Algorithm 1:** Counting Size-3 $(d, s)$-Graphlets

**Input:** (1) $d$-Graph $G^{(d)} = (V, E^{(\leq d)})$ of graph $G$
   (2) $d$-DAG $\vec{G}^{(d)} = (V, \vec{E}^{(\leq d)})$ of graph $G$
   (3) Maximum considered distance $d$

**Output:** The count of each size-3 $(d, s)$-graphlet $T_i^{(d)}$:
   $C(T_i^{(d)}) \;\; \forall i \in \{1, \cdots, |T^{(d)}|\}$

 // Initialization

1   $C(T_i^{(d)}) \leftarrow 0 \;\; \forall i \in \{1, \cdots, |T^{(d)}|\}$

 // Count non-deducible $(d, s)$-graphlets $\overline{T}^{(d)}$

2   **for each** $u \in V$

3     $P_u \leftarrow$ EFFECTIVE_NEIGHBOR_PAIRS$(u, d, \vec{G}^{(d)})$

4     **for each** $(v, w) \in P_u$

5       $T_*^{(d)} \leftarrow$ GET_TRIANGLE$\big((u, v, w), d, \vec{G}^{(d)}\big)$

6       $C(T_*^{(d)}) \leftarrow C(T_*^{(d)}) + 1$

 // Count deducible $(d, s)$-graphlets $\widehat{T}^{(d)}$

7   **for each** $T_j^{(d)} \in \widehat{T}^{(d)}$

8     $C(T_j^{(d)}) \leftarrow$ COMB_THREE$\Big(T_j^{(d)}, \{C(T_i^{(d)})\}_{i=1}^{|T^{(d)}|}, G^{(d)}\Big)$

9   **return** $C(T_i^{(d)}) \;\; \forall i \in \{1, \cdots, |T^{(d)}|\}$

---

$(T^{(3)})$. As detailed in Algorithm 1 [1], we categorize size-3 $(d, s)$-graphlets $(T^{(d)})$ into two groups: *non-deducible* and *deducible* $(d, s)$-graphlets as follows:

- **Non-deducible size-3 $(d, s)$-graphlets** $(\overline{T}^{(d)})$ require explicit enumeration, as their counts cannot be directly inferred. These include the following types of triangles:
  - $\overline{T}^{(2)} = \{T_1^{(2)}, T_2^{(2)}, T_4^{(2)}\}$
  - $\overline{T}^{(3)} = \{T_1^{(3)}, T_2^{(3)}, T_3^{(3)}, T_4^{(3)}, T_6^{(3)}, T_7^{(3)}, T_9^{(3)}\}$
- **Deducible size-3 $(d, s)$-graphlets** $(\widehat{T}^{(d)})$ are those whose counts can be inferred from the graph structure (e.g., node degrees) and the counts of non-deducible $(d, s)$-graphlets:
  - $\widehat{T}^{(2)} = \{T_3^{(2)}, T_5^{(2)}, T_6^{(2)}\}$
  - $\widehat{T}^{(3)} = \{T_5^{(3)}, T_8^{(3)}, T_{10}^{(3)}, T_{11}^{(3)}, T_{12}^{(3)}, T_{13}^{(3)}\}$

As we describe in detail below, we first selectively enumerate each instance of non-deducible $(d, s)$-graphlets (lines 2 - 6). Afterward, the counts of deducible $(d, s)$-graphlets can be rapidly computed using specialized combinatorial methods without enumeration (lines 7 - 8). This adaptive counting scheme significantly improves the speed of EDGE, as empirically demonstrated in Section 6.

**Counting non-deducible $(d, s)$-graphlets.** To count each non-deducible $(d, s)$-graphlet, EDGE iterates over each node $u$. It samples a subset of its neighboring pairs to ensure that only instances of non-deducible $(d, s)$-graphlets are enumerated (line 3). For each *effective* neighboring pair $(v, w)$, it identifies the $(d, s)$-graphlet of the triangle $(u, v, w)$ based on the distances between the constituent nodes (line 5). The count of the corresponding $(d, s)$-graphlet is then incremented (line 6).

**Counting deducible $(d, s)$-graphlets.** Once EDGE counts the non-deducible $(d, s)$-graphlets, it efficiently computes the counts of deducible $(d, s)$-graphlets using combinatorial counting (line 8). For

---

[1]Refer to Appendix A.3 for details on its sub-algorithms.

---

each deducible $(d, s)$-graphlet, EDGE leverages predefined equations specific to each $(d, s)$-graphlet, based on (1) the exact count of the non-deducible $(d, s)$-graphlets and (2) structural information (e.g., node degree), if needed. This deductive approach avoids the need for explicit enumeration for deducible $(d, s)$-graphlets. For example, $C(T_5^{(2)})$ can be computed by using the following equation:

$$C(T_5^{(2)}) = \sum_{u \in V} \binom{|N_u^{(2)}|}{2} - 3C(T_1^{(2)}) - C(T_2^{(2)})$$

The first term counts all cases where the center node of $T_5^{(2)}$ has neighbors connected by 2-edges on both sides. Since these neighbors may also be connected, the counts of the non-deducible $(2, 3)$-graphlets $T_1^{(2)}$ and $T_2^{(2)}$, are subtracted. As $T_1^{(2)}$ can appear at any of the three nodes in a triangle, its count is multiplied by 3 when subtracting. While we do not detail the derivation of equations for all deducible $(d, s)$-graphlets, we experimentally verified the correctness. All specific equations for COMB_THREE are provided in Algorithm 7 of Appendix A.3.

**Complexity analysis.** We analyze the time complexity of EDGE for counting size-3 $(d, s)$-graphlets (Algorithm 1) in Theorem 1.

**THEOREM 1 (COMPLEXITY OF ALGORITHM 1).** *The time complexity of EDGE for counting size-3 $(d, s)$-graphlets is $O(|V| d^4 \vec{\Delta}^{2d} \log \vec{\Delta})$, where $\vec{\Delta}$ is the maximum out-going node degree, i.e., $\vec{\Delta} = \max_{u \in V} |\vec{N}_u^{(1)}|$.*

**PROOF.** Refer to Appendix A.3.

**Remarks.** The time complexity of Algorithm 1 is primarily dominated by the counting of only non-deducible $(d, s)$-graphlets. EDGE achieves the complexity in two ways: (1) It employs $d$-DAGs, where each node has fewer neighbors compared to $d$-graphs, reducing redundancy in enumeration. (2) EDGE selectively enumerates only non-deducible $(d, s)$-graphlets and rapidly counts deducible $(d, s)$-graphlets afterward. As demonstrated in Section 6, these optimizations lead to a significant speedup of EDGE.

## 5.3 Size-4 $(d, s)$-Graphlet Counting

We now describe how EDGE counts size-4 $(d, s)$-graphlets (i.e., $s = 4$), focusing on $(2, 4)$-graphlets $(Q^{(2)})$, as outlined in Algorithm 2 [2]. For $(2, 4)$-graphlets, we categorize the 36 possible configurations (i.e., $Q_1^{(2)}$ - $Q_{36}^{(2)}$) into *non-deducible*, *semi-deducible*, and *deducible*.

- **Non-deducible $(2, 4)$-graphlets** $(\overline{Q}^{(2)})$, which are all *cliques*, require explicit enumeration to obtain their exact counts:
  - $\overline{Q}^{(2)} = \{Q_1^{(2)}, Q_2^{(2)}, \cdots, Q_{11}^{(2)}\}$
- **Semi-deducible $(2, 4)$-graphlets** $(\widetilde{Q}^{(2)})$, which are all *cycles*, partially require enumeration, and their counts are then adjusted:
  - $\widetilde{Q}^{(2)} = \{Q_{28}^{(2)}, Q_{29}^{(2)}, Q_{30}^{(2)}\}$
- **Deducible $(2, 4)$-graphlets** $(\widehat{Q}^{(2)})$ are those whose counts can be rapidly obtained using the counts of non-deducible $(2, 4)$-graphlets and the graph structure:
  - $\widehat{Q}^{(2)} = \{Q_{12}^{(2)}, \cdots, Q_{27}^{(2)}, Q_{31}^{(2)}, \cdots, Q_{36}^{(2)}\}$

We first count the non-deducible $(d, s)$-graphlets by enumerating over the graph (lines 3 - 10) and use their counts to compute the counts of the deducible ones through combinatorial methods

---

[2]Refer to Appendix A.4 for details on its sub-algorithms.

---

**Algorithm 2:** $(2,4)$-Graphlets Counting

**Input:** (1) 2-graph $G^{(2)} = (V, E^{(\leq 2)})$ of graph $G$

(2) 2-DAG $\vec{G}^{(2)} = (V, \vec{E}^{(\leq 2)})$ of graph $G$

**Output:** The count of each size-4 $(d,s)$-Graphlets $Q_i^{(2)}$:

$$C(Q_i^{(2)}) \quad \forall i \in \{1, \cdots, |Q^{(2)}|\}$$

// Initialization

1 $C(Q_i^{(2)}) \leftarrow 0 \quad \forall i \in \{1, \cdots, |Q^{(2)}|\}$

2 **for each** $u \in V$

      // Count non-deducible $\overline{Q}^{(2)}$

3     **for each** $v \in \vec{N}_u^{(1)} \cup \vec{N}_u^{(2)}$

4         $N_{u,v} \leftarrow \left( \vec{N}_u^{(1)} \cup \vec{N}_u^{(2)} \right) \cap \left( \vec{N}_v^{(1)} \cup \vec{N}_v^{(2)} \right)$

5         $\mathcal{T}_{u,v} \leftarrow \text{Triangle\_Pairs}((u,v), N_{u,v}, \vec{G}^{(2)})$

6         **for each** $((u,v,w),(u,v,w')) \in \mathcal{T}_{u,v}$

7             $T_\circ^{(2)} \leftarrow \text{Get\_Triangle}\left((u,v,w), \vec{G}^{(2)}\right)$

8             $T_\bullet^{(2)} \leftarrow \text{Get\_Triangle}\left((u,v,w'), \vec{G}^{(2)}\right)$

9             $Q_*^{(2)} \leftarrow \text{Get\_Clique}\left(T_\circ^{(2)}, T_\bullet^{(2)}, u, v, w, w'\right)$

10            $C(Q_*^{(2)}) \leftarrow C(Q_*^{(2)}) + 1$

      // Count semi-deducible $\widetilde{Q}^{(2)}$

11     **for each** $(v,v') \in (\vec{N}_u^{(1)} \cup \vec{N}_u^{(2)}) \times \vec{N}_u^{(2)}$

12         **for each** $w \in \{w' \in N_v^{(2)} \cap \left(N_{v'}^{(1)} \cup N_{v'}^{(2)}\right) : u \prec^{(d)} w'\}$

13             $T_\triangle^{(2)} \leftarrow \text{Get\_Non-Induced\_Wedge}\left((u,v,w), \vec{G}^{(2)}\right)$

14             $T_\blacktriangle^{(2)} \leftarrow \text{Get\_Non-Induced\_Wedge}\left((u,v',w), \vec{G}^{(2)}\right)$

15             $Q_*^{(2)} \leftarrow \text{Get\_Non-Induced\_Cycle}\left(T_\triangle^{(2)}, T_\blacktriangle^{(2)}\right)$

16            $C(Q_*^{(2)}) \leftarrow C(Q_*^{(2)}) + 1$

    // Count deducible $(d,s)$-graphlets $\widehat{Q}^{(2)}$

17 **for each** $Q_j^{(2)} \in \widehat{Q}^{(2)}$

18     $C(Q_j^{(2)}) \leftarrow \text{Comb\_Four}\left(Q_j^{(2)}, \{C(Q_i^{(2)})\}_{i=1}^{|Q^{(2)}|}, G^{(2)}\right)$

    // Adjust counts of semi-deducible $(d,s)$-graphlets $\widetilde{Q}^{(2)}$

19 **for each** $Q_j^{(2)} \in \widetilde{Q}^{(2)}$

20     $C(Q_j^{(2)}) \leftarrow \text{Comb\_Four}\left(Q_j^{(2)}, \{C(Q_i^{(2)})\}_{i=1}^{|Q^{(2)}|}\right)$

21 **return** $C(\overline{Q}_i^{(2)}) \quad \forall i \in \{1, \cdots, |\overline{Q}^{(2)}|\}$

---

(lines 17 - 18). For semi-deducible ones, we initially compute the number of their non-induced instances (i.e., instances that induce semi-deducible $(d,s)$-graphlets) (lines 11 - 16) and adjust their counts accordingly after the enumeration (lines 19 - 20).

**Counting non-deducible $(d,s)$-graphlets.** Instead of exhaustively enumerating all size-4 instances in the graph to obtain exact count of $(2,4)$-graphlets, we count them by decomposing their structure. Notably, all non-deducible $(d,s)$-graphlets form cliques, which can be decomposed into two triangles sharing an edge, and the remaining two nodes are also connected (see an example in Figure 3). For every edge $(u,v)$, using their common neighbors (line 4), we determine the set of triangle pairs where the remaining nodes are connected (line 5). For each pair of triangles, we first identify the $(2,3)$-graphlet of each triangle (lines 7 - 8) and then determine the $(2,4)$-graphlet based on the combination of the two $(2,3)$-graphlets

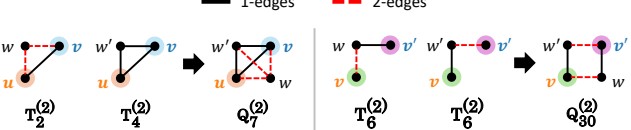

**Figure 3: Examples of: (left) two triangles forming a clique, assuming $w$ and $w'$ are connected by a 2-edge, and (right) two wedges forming a cycle, assuming $w$ and $w'$ are disconnected.**

(line 9). The count of the identified $(2,4)$-graphlet is then incremented (line 10).

**Counting semi-deducible $(d,s)$-graphlets.** For semi-deducible $(2,4)$-graphlets, which are cycles, we adopt a two-step approach. Note that a cycle is composed of two wedges that share two end nodes (see an example in Figure 3). Based on this structure, given two (dis)connected nodes $(v,v')$, we first enumerate pairs of non-induced wedges (which can be either wedges or triangles) using their common neighbors. Next, we identify the $(2,3)$-graphlet of each non-induced wedge (lines 13 - 14). Based on their combination, we determine the $(2,4)$-graphlet of the non-induced cycle (line 15) and increment the count (line 16). Once the enumeration finishes, we adjust the counts by subtracting the counts of $(2,4)$-graphlets that are not cycles, ensuring the correct count of semi-deducible $(d,s)$-graphlets (lines 19 - 20).

**Counting deducible $(d,s)$-graphlets.** The counts of deducible $(2,4)$-graphlets can be rapidly computed from the counts of the non-deducible $(2,4)$-graphlets (lines 17 - 18). The equations of Comb\_Four are provided in Algorithm 13 of Appendix A.4.

**Complexity analysis.** We analyze the time complexity of EDGE for counting $(2,4)$-graphlets (Algorithm 2) in Theorem 2.

**THEOREM 2 (COMPLEXITY OF ALGORITHM 2).** *The time complexity of EDGE for counting $(2,4)$-graphlets is $(|V|\vec{\Delta}^4\Delta^2 \log \vec{\Delta})$, where $\Delta$ and $\vec{\Delta}$ denote the maximum undirected degree and the maximum outgoing degree, respectively, i.e., $\Delta = \max_{u \in V} |N_u^{(1)}|$ and $\vec{\Delta} = \max_{u \in V} |\vec{N}_u^{(1)}|$.*

**PROOF.** Refer to Appendix A.4.

## 6 EXPERIMENTS

We share our empirical analysis using $(d,s)$-graphlets and its counting algorithm, EDGE. We aim to answer the following questions:

- **Q1. Graph characterization:** How effective are $(d,s)$-graphlets in distinguishing and clustering graphs across different domains?
- **Q2. Real-world discoveries:** What insights do $(d,s)$-graphlets provide that cannot be uncovered by simple graphlets?
- **Q3. Speed and scalability:** How fast and scalable is EDGE? Do $d$-DAG and deduced counting contribute to its efficiency?

### 6.1 Experimental Settings

We report the settings where the experiments were performed.

**Datasets.** We used 13 real-world graphs from five different domains: collaboration [3, 13, 25, 34], web [11, 12], social-Facebook [59, 60], tags [7], and road [36]. We present basic statistics of the datasets with further details in Appendix B.1.

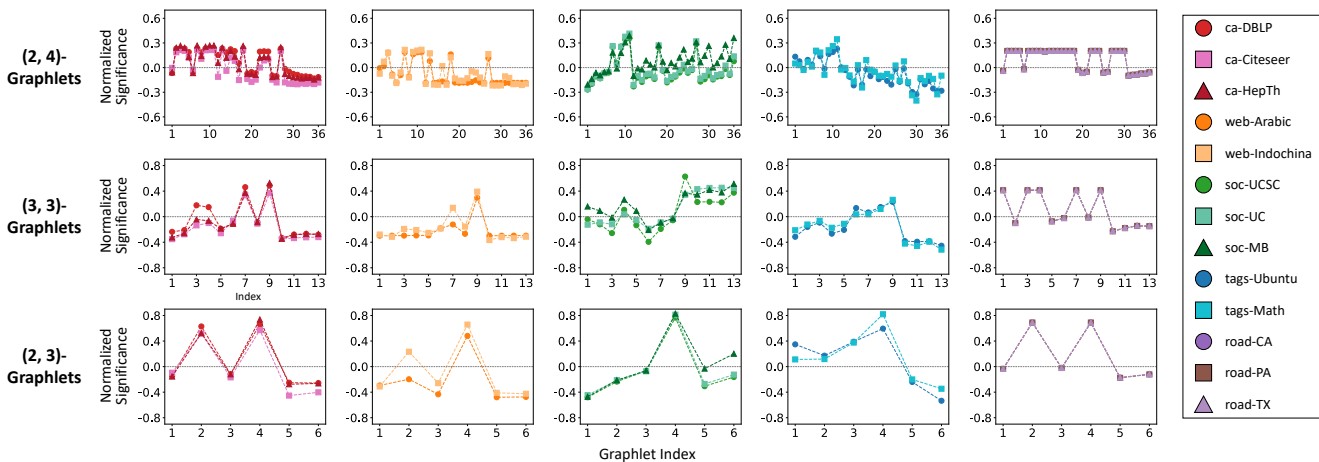

**Figure 4: Graphs from the same domain exhibit similar CPs derived from the counts of $(2, 4)$-, $(3, 3)$-, and $(2, 3)$-graphlets.**

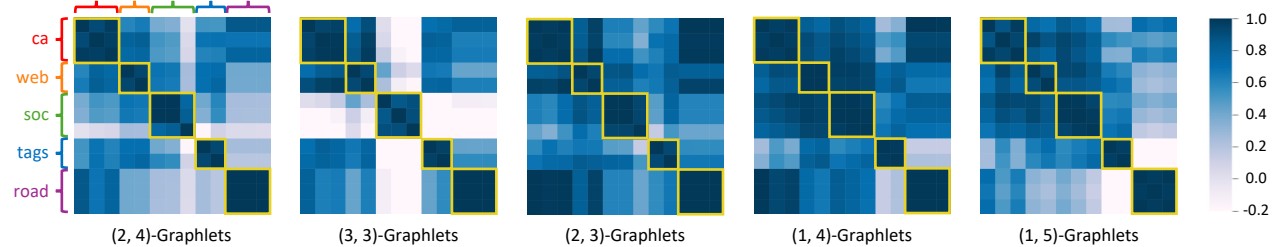

**Figure 5: The domains of the graphs are effectively distinguished by CPs derived from the counts of $(d, s)$-graphlets. Specifically, $(2, 4)$-, $(3, 3)$-, and $(2, 3)$-graphlets, which account for generalized distances, provide a clearer distinction of graphs across domains compared to simple graphlets that consider only direct connections (e.g., $(1, 4)$- and $(1, 5)$-graphlets, which represent size-4 and size-5 graphlets, respectively). For numerical comparisons, refer to Table 2.**

**Table 2:** $(d, s)$-graphlets exhibit larger correlation gaps between graphs within the same domain and across domains, as well as superior clustering performance compared to the original graphlets.

|  | $(d, s)$ | Correlation Gap (Within - Across) | Clustering F1 | NMI | SH |
|---|---|---|---|---|---|
| Original | $(1, 3)$ | 0.000 | 0.467 | 0.455 | 0.266 |
|  | $(1, 4)$ | 0.252 | 0.670 | 0.772 | 0.539 |
|  | $(1, 5)$ | 0.440 | 0.920 | 0.908 | 0.680 |
| Ours | $(2, 3)$ | 0.253 | 0.667 | 0.856 | 0.585 |
|  | $(3, 3)$ | **0.667** | **1.000** | **1.000** | **0.797** |
|  | $(2, 4)$ | 0.473 | **1.000** | **1.000** | 0.795 |

**Implementations.** We implemented EDGE in C++. EDGE supports multi-threading, and we set the number of threads to 6. To count size-4 and size-5 graphlets, we used the open-source C++ implementations of recent exact counting methods, PGD [46] and ESCAPE [23]. We ran PGD with 6 threads, while ESCAPE does not support multi-threading.

**Machines.** We performed all the experiments on a machine with an Intel i9-10900K CPU and 64GB memory.

## 6.2 Q1. Graph Characterization

We analyzed the characterization power of the $(d, s)$-graphlets counted by EDGE. Specifically, we computed the characteristic profiles (CPs; see Section 4.3) for each graph using counts of $(2, 3)$, $(2, 4)$, and $(3, 3)$-graphlets. As shown in Figure 4, graphs within the same domain exhibit highly similar CPs for all $(d, s)$-graphlets, while CPs are clearly distinguished across different domains.

We computed the correlations between CPs of different graphs, as shown in Figure 5. Notably, $(2, 4)$-, $(3, 3)$-, and $(2, 3)$-graphlets provide clearer distinctions between graphs across domains compared to the original graphlets (i.e., $(1, 4)$- and $(1, 5)$-graphlets). Specifically, as shown in Table 2, the correlation gap (i.e., the difference between average correlations within and across domains) is largest for $(3, 3)$-graphlets, followed by $(2, 4)$-graphlets, even though they use fewer nodes per graphlet than $(1, 5)$-graphlets. These large gaps demonstrate the effectiveness of incorporating multi-hop distances for graphlets.

We further evaluated the clustering of graphs using the CPs as input features, specifically applying spectral clustering. As shown in Table 2, $(d, s)$-graphlets lead to higher clustering performance in terms of F1 score, NMI, and Silhouette score. This further validates the effectiveness of $(d, s)$-graphlets in graph characterization.

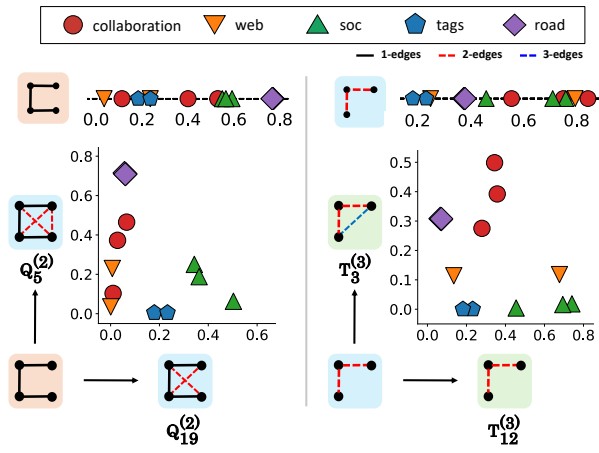

**Figure 6: Incorporating higher-order node distances allows for finer differentiation of local structures. (Top) The ratio of a graphlet relative to the total number of instances. (Bottom) The graphlet can be further decomposed into finer structures when considering distances beyond those in the graphlet.**

## 6.3 Q2. Real-World Discoveries

We conduct case studies on real-world data that support the granularity and generality of $(d, s)$-graphlets.

**Finer granularity.** We first demonstrate that incorporating higher-order distances between nodes allows for finer differentiation of local structures, as shown in Figure 6. On the left, the original 3-path size-4 graphlet is divided into two cases, $Q_5^{(2)}$ and $Q_{19}^{(2)}$, when considering distances up to 2. A 2D representation shows that the proportions of these finer local structures lead to clearer graph characterization. For example, graphs in the collaboration domain, were indistinguishable when considering only the 3-path graphlet, become distinguishable due to a lower proportion of $Q_{19}^{(2)}$. Similarly, on the right, a $(2, 3)$-graphlet is generalized to $(3, 3)$-graphlets, where $T_5^{(2)}$ splits into $T_3^{(2)}$ and $T_{12}^{(2)}$ when considering distances up to 3. This further improves domain distinction and thus enhanced graph characterization.

**Comprehensiveness.** To assess the significance of the $(d, s)$-graphlets in graph characterization, we use a scoring function that measures each $(d, s)$-graphlet's contribution to distinguishing graphs by domain [32]. Based on these scores, we ranked the $(d, s)$-graphlets and retrieved the top 5 from each graph. Notably, 59 out of the 65 retrieved $(d, s)$-graphlets were newly defined $(2, 4)$-graphlets, which cannot be described by simple graphlets, demonstrating the effectiveness of $(d, s)$-graphlets in capturing more comprehensive structural patterns. For more details, refer to Appendix B.2.

## 6.4 Q3. Speed and Scalability

We evaluate the speed and scalability of EDGE by comparing it with its variants and existing counting methods.

**Effects of EDGE's components.** We first analyze the effects of EDGE's design components: (1) deducible counting for deducible $(d, s)$-graphlets (i.e., $\widetilde{T}^{(d)}$ and $\widetilde{Q}^{(d)}$) and (2) using $d$-DAG for counting non- and semi-deducible $(d, s)$-graphlets. We evaluate two variants: EDGE-D2, which removes both (1) and (2), and EDGE-D, which removes only (1). As shown in Figure 7, EDGE is significantly faster

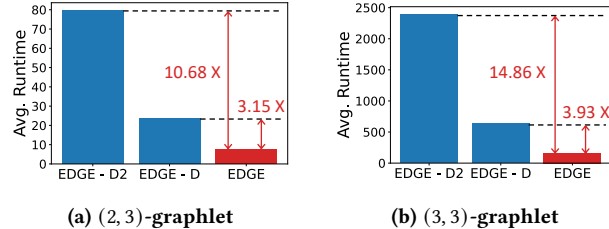

(a) $(2, 3)$-graphlet          (b) $(3, 3)$-graphlet

**Figure 7: EDGE is faster than its variants: (1) EDGE-D2, which lacks deducible counting and $d$-DAG, and (2) EDGE-D, which lacks $d$-DAG. This demonstrates the effectiveness of EDGE's design choices for fast $(d, s)$-graphlet counting.**

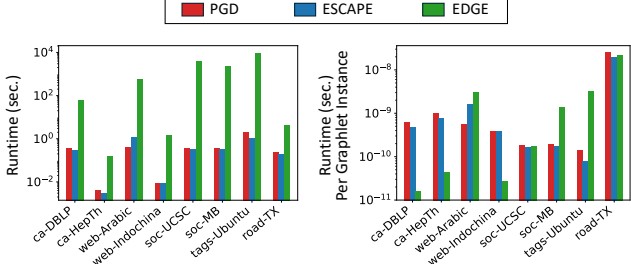

**Figure 8: Comparison between EDGE (for counting $(2, 4)$-graphlets) and PGD & ESCAPE (for counting $(1, 4)$-graphlets). (Left) In total runtime, EDGE is slower than PGD and ESCAPE due to the additional virtual edges. (Right) In runtime per instance, EDGE is competitive and even faster in some cases.**

than these variants, achieving up to 14.86× and 3.93× speedups over EDGE-D2 and EDGE-D, respectively. These results demonstrate the effectiveness of EDGE's design choices in avoiding unnecessary enumeration and using $d$-DAGs to reduce the neighbors. For more details, see Appendix B.3.

**Comparison to graphlet counting methods.** We compare the counting times of PGD and ESCAPE for $(1, 4)$-graphlets with EDGE for $(2, 4)$-graphlets. As shown in Figure 8, EDGE is generally slower than PGD and ESCAPE due to the additional edges connecting indirect nodes. However, when comparing per graphlet, EDGE's runtime is competitive, and in some cases, even faster than the baselines. For more details, see Appendix B.3.

## 7 CONCLUSIONS

We present $(d, s)$-graphlets, distance-generalized graphlets for enhanced graph characterization. Our contributions are as follows:

- **New Definition:** We formally define $(d, s)$-graphlets, distance-generalized graphlets that provide more detailed analysis of local structures (Section 4).
- **Efficient Counting Algorithm:** We introduce EDGE, an optimized counting algorithm for $(d, s)$-graphlets that reduces unnecessary enumeration through deducible computation and efficient data structures (Section 5).
- **Comprehensive Experiments:** Our experiments across 13 real-world datasets demonstrate both the effectiveness of $(d, s)$-graphlets and the efficiency of EDGE (Section 6).

For reproducibility, our code and datasets are available at [26]. Future research directions include accelerating the counting of $(d, s)$-graphlets through approximation.

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

# A ALGORITHM DETAILS

In this section, we provide a more detailed explanation of the counting algorithms introduced in Section 5.

## A.1 $d$-Graph Construction

To count the instances of $(d, s)$-graphlets, we construct $E^{(\leq d)} = \{E^{(1)}, E^{(2)}, \cdots, E^{(d)}\}$ as a preprocessing step. Algorithm 3 describes the process of constructing $d$-edges (i.e., $E^{(d)}$) from the input graph. For each node $u \in V$, we utilize the BFS function to identify all nodes that are exactly $d$-hops away from $u$ (line 3). Then, these nodes are added as edges in $E^{(d)}$ (line 6) while avoiding duplicates. The time complexity of this preprocessing step is given in Lemma 1, and we provide its proof as follows:

**PROOF OF LEMMA 1.** For each node $u \in V$, the algorithm performs a BFS up to $d$-hops, and thus the total number of nodes explored up to $d$-hops is $O(\Delta + \Delta^2 + \cdots + \Delta^d) = O(\Delta^d)$. Thus, the time complexity for performing for all nodes $u \in V$ is $O(|V|\Delta^d)$.

## A.2 Effects of $d$-DAG

In this section, we assess the effectiveness of using $d$-DAGs instead of (undirected) $d$-graphs. As shown in Table 3, both the average and maximum degrees of nodes with respect to 1-edges, 2-edges, and 3-edges are significantly smaller in $d$-DAGs compared to $d$-graphs, i.e., $d_{avg} \gg \vec{d}_{avg}$ and $\Delta \gg \vec{\Delta}$. This reduction dramatically decreases the number of enumerations required for counting non-deducible $(d, s)$-graphlets. We empirically demonstrate the effectiveness of employing $d$-DAGs in Section 6.

---

**Algorithm 3:** $d$-Edge Construction (Preprocess)

**Input:** (1) Input graph $G = (V, E = E^{(1)})$
       (2) Maximum distance considered $d$
**Output:** Set of $d$-edges $E^{(d)}$

1   $E^{(d)} \leftarrow \emptyset$
2   **for** *each* $u \in V$
3      // Get all nodes at exactly $d$-hops from $u$
      $S^{(d)} \leftarrow \text{BFS}(u, d, G)$
4      **for** *each* $v \in S^{(d)}$
5         **if** $u \prec v$
6           $E^{(d)} \leftarrow E^{(d)} \cup \{(u, v)\}$
7   **return** $E^{(d)}$

---

**Table 3: Degree statistics for various datasets. Each value represents the degree characteristics of 1-edge, 2-edge, and 3-edge types. $\Delta$ represents the maximum degree, $\vec{\Delta}$ denotes the out-going maximum degree, and $d_{avg}$ and $\vec{d}_{avg}$ represent the average degree and the out-going average degree, respectively.**

| Datasets | Edge type | $d_{avg}$ | $\Delta$ | $\vec{d}_{avg}$ | $\vec{\Delta}$ |
|---|---|---|---|---|---|
| ca-DBLP | 1-edge | 3 | 343 | 2 | 113 |
| | 2-edge | 39 | 5.15K | 20 | 378 |
| | 3-edge | 482 | 42.9K | 241 | 3.10K |
| ca-Citeseer | 1-edge | 3 | 1.37K | 2 | 206 |
| | 2-edge | 32 | 5.45K | 16 | 1.36K |
| | 3-edge | 221 | 24.4K | 111 | 1.91K |
| web-Arabic | 1-edge | 10 | 1.10K | 5 | 164 |
| | 2-edge | 18 | 2.66K | 9 | 1.10K |
| | 3-edge | 91 | 15.8K | 46 | 1.13K |
| web-Indochina | 1-edge | 4 | 199 | 2 | 83 |
| | 2-edge | 37 | 2.01K | 19 | 194 |
| | 3-edge | 325 | 3.92K | 163 | 1.47K |
| soc-UCSC | 1-edge | 24 | 454 | 12 | 82 |
| | 2-edge | 799 | 5.64K | 400 | 1.46K |
| | 3-edge | 2.76K | 6.91K | 1.38K | 4.51K |
| soc-UC | 1-edge | 22 | 660 | 11 | 99 |
| | 2-edge | 690 | 4.90K | 345 | 1.35K |
| | 3-edge | 1.97K | 5.12K | 987 | 3.19K |

## A.3 Details of Algorithm 1

In this subsection, we provide the details of sub-algorithms used in Algorithm 1. Then, we provide the proof of Theorem 1 regarding the complexity of Algorithm 1.

**RETRIEVE_DISTANCE (Algorithm 4).** Given a pair of nodes $(u, v)$ and the $d$-DAG $\vec{G}^{(d)}$, which considers up to $d$-edges, this function computes the distance between $u$ and $v$. Equivalently, it determines in which of $\vec{E}^{(\leq d)} = \{\vec{E}^{(1)}, \cdots, \vec{E}^{(d)}\}$ the edge $(u, v)$ is contained. If $(u, v)$ is not included in any of the $\vec{E}^{(1)}, \cdots, \vec{E}^{(d)}$, the distance is considered to be $\infty$. Since the $d$-DAG is $d$-degree-ordered, a directed edge $u \rightarrow v$ (or $v \rightarrow u$) exists if $u \prec^{(d)} v$ (or $v \prec^{(d)} u$). For each distance $d' \in \{1, \cdots, d\}$, it checks whether $v$ is an out-going neighbor of $u$ (i.e., $v \in \vec{N}_u^{(d')}$) if $u \prec^{(d)} v$, and vice versa. If the neighbor is found, the function returns the corresponding distance $d'$. The time complexity of Algorithm 4 is provided in Lemma 2.

**Algorithm 4:** Retrieve_Distance

**Input:** (1) A pair of nodes $u$ and $v$
       (2) $d$-DAG $\vec{G}^{(d)} = (V, \vec{E}^{(\leq d)})$ of graph $G$
       (3) Maximum considered distance $d$
**Output:** $\delta(u, v)$ (distance between $u$ and $v$)

1   **for each** $d' \in \{1, \cdots, d\}$
2      **if** $u \prec^{(d)} v$
3         **if** $v \in \vec{N}_u^{(d')}$                  ▷ Binary Search
4            **return** $d'$
5      **else**
6         **if** $u \in \vec{N}_v^{(d')}$                  ▷ Binary Search
7            **return** $d'$

8   **return** $\infty$

**Lemma** 2 (Complexity of Algorithm 4). *The time complexity of retrieving the distance between two nodes is $O(d^2 \log \vec{\Delta})$, where $\vec{\Delta}$ is the maximum out-degree, i.e., $\vec{\Delta} = \max_{u \in V} |\vec{N}_u^{(1)}|$.*

**Proof.** Without loss of generality, let $u \prec^{(d)} v$. For each distance $d' \in \{1, \cdots, d\}$, we check $v$ is the out-going $d'$-neighbor of $u$, i.e., $v \in \vec{N}_u^{(d')}$. Assuming that $\vec{N}_u^{(d')}$ is implemented as a sorted list, we employ a binary search with a time complexity of $O(\log |\vec{N}_u^{(d')}|)$. In the worst case, we search through all distances $d' \in \{1, \cdots, d\}$, yielding a total time complexity of $O(\log |\vec{N}_u^{(1)}| + \cdots + \log |\vec{N}_u^{(d)}|)$. Let $\vec{\Delta}$ be the maximum out-degree, i.e., $\vec{\Delta} = \max_{v \in V} |\vec{N}_v^{(1)}|$. The time complexity of Algorithm 4 is thus derived as $O(\log \vec{\Delta} + \cdots \log \vec{\Delta}^d) = O(d^2 \log \vec{\Delta})$.

**Effective_Neighbor_Pairs (Algorithm 5).** This function determines the *effective* neighboring pairs from all possible pairs of neighbors. Specifically, given a node $u$ and the maximum considered distance $d$, the set of all neighbors of node $u$ up to distance $d$ is $\bigcup_{d' \in \{1, \cdots, d\}} \vec{N}_u^{(d')}$. Among all pairs of neighbors (i.e., $\binom{\bigcup_{d' \in \{1, \cdots, d\}} \vec{N}_u^{(d')}}{2}$), we only consider those that are connected and thus form a triangle with $u$. In addition, we exclude the neighbor pairs whose triangle forms deducible $(d, s)$-graphlets, as these can be efficiently counted without enumeration. This reduction in the set of neighbor pairs significantly and speeds up EDGE. The time complexity of Algorithm 5 is provided in Lemma 3.

**Lemma** 3 (Complexity of Algorithm 5). *The time complexity of retrieving effective neighboring pairs of a node is $O(d^4 \vec{\Delta}^{2d} \log \vec{\Delta})$, where $\vec{\Delta}$ is the maximum out-degree, i.e., $\vec{\Delta} = \max_{u \in V} |\vec{N}_u^{(1)}|$.*

**Proof.** For a given node $u$ and the maximum considered distance $d$, we consider $u$'s out-going neighbors at all $O(d^2)$ combinations of distances $(d_i, d_j)$. For each pair of distances, we examine all pairs $(v, w)$ (where $v \prec^{(d)} w$) of $d_i$-neighbors and $d_j$-neighbors, which results is $O(|\vec{N}_u^{d_i}| \cdot |\vec{N}_u^{d_j}|) = O(\vec{\Delta}^{2d})$ pairs, where $\vec{\Delta}$ is the maximum out-degree, i.e., $\max_{u \in V} |\vec{N}_u^{(1)}|$. Next, we check whether the two neighbors are connected, which can be done by performing a binary search for $w$ from $\vec{N}_v^{(d_k)}$ $\forall d_k \in \{1, \cdots, d_k\}$ which takes $O(\log |\vec{N}_v^{(1)}| + \cdots + \log |\vec{N}_v^{(d)}|) = (d^2 \log \vec{\Delta})$ time. Thus, the overall time complexity is $O(d^4 \vec{\Delta}^{2d} \log \vec{\Delta})$.

**Algorithm 5:** Effective_Neighbor_Pairs

**Input:** (1) Node $u$
       (2) Maximum considered distance $d$
       (3) $d$-DAG $\vec{G}^{(d)} = (V, \vec{E}^{(\leq d)})$ of graph $G$
**Output:** Effective node pairs set $P_u$

1   $P_u \leftarrow \emptyset$
2   **for each** $(d_i, d_j) \in \{(d_i', d_j') : 1 \leq d_i' \leq d_j' \leq d\}$
3      **for each** $(v, w) \in \{(v', w') \in \vec{N}_u^{(d_i)} \times \vec{N}_u^{(d_j)} : v' \prec^{(d)} w'\}$
4         **if** $w \in \vec{N}_v^{(d_k)} \exists d_k \in \{1, \cdots, d\} \setminus \{d_i + d_j, |d_i - d_j|\}$
5            $P_u \leftarrow P_u \cup \{(v, w)\}$

6   **return** $P_u$

**Get_Triangle (Algorithm 6).** Given three nodes $(u, v, w)$ which consists a triangle, and the maximum distance considered $d$, this function returns the corresponding $(d, 3)$-graphlet of the triangle. First, it retrieves the distances for all pairs of edges. Then, based on these three distances, it identifies the $(d, 3)$-graphlet. The time complexity of Algorithm 6 is provided in Lemma 4.

**Lemma** 4 (Complexity of Algorithm 6). *The time complexity of identifying $(d, 3)$-graphlet of a triangle is $O(d^2 \log \vec{\Delta})$, where $\vec{\Delta}$ is the maximum out-degree, i.e., $\vec{\Delta} = \max_{u \in V} |\vec{N}_u^{(1)}|$.*

**Proof.** The time complexity for retrieving the distances of three pairs of nodes is $O(d^2 \log \vec{\Delta})$ (from Lemma 2). Once the distances are obtained, the corresponding $(d, 3)$-graphlet can be identified in $O(1)$ time. Thus, the overall time complexity is $O(d^2 \log \vec{\Delta})$.

**Comb_Three (Algorithm 7).** This function is for computing the counts of deducible $(d, 3)$-graphlets ($\widehat{T}^{(d)}$). Based on the counts of non-deducible $(d, 3)$-graphlets ($\bar{T}^{(d)}$), it computes the counts of the target deducible $(d, 3)$-graphlets. The time complexity of Algorithm 7 is provided in Lemma 5.

**Lemma** 5 (Complexity of Algorithm 7). *The time complexity of computing the count of the given deducible $(d, 3)$-graphlet is $O(|V|)$.*

**Proof.** To compute the count of the target deducible $(d, 3)$-graphlet, degree-based computations are required, e.g., $\sum_{u \in V} \binom{|N_u^{(1)}|}{2}$ to compute $C(T_3^{(2)})$. This takes $O(|V|)$ time.

**Proof of Theorem 1.** Below, we provide the proof for Theorem 1.

**Proof.** The time complexity of Algorithm 1 is determined by two main operations: (1) counting non-deducible $(d, 3)$-graphlets and (2) counting deducible $(d, 3)$-graphlets.

- To count **non-deducible** $(d, 3)$-graphlets, we iterate over each node $u \in V$ and obtain its effective neighbor pairs using Effective_Neighbor_Pairs which takes $O(d^4 \vec{\Delta}^{2d} \log \Delta)$ time (Lemma 3). For each effective neighbor pair $(v, w)$, we identifies the $(d, 3)$-graphlet of the triangle $(u, v, w)$ using Get_Triangle, which takes $O(d^2 \log \vec{\Delta})$ time (Lemma 4). In practice, we (1) check the connectivity between $v$ and $w$ takes $O(d^2 \log \vec{\Delta})$ time, and (2) subsequently identify the $(d, 3)$-graphlet if it is connected which takes $O(|V| d^2 \log \vec{\Delta})$ time as well. Thus, the total time complexity of counting non-deducible $(d, 3)$-graphlets is $O(d^4 \vec{\Delta}^{2d} \log \vec{\Delta})$.
- To count **deducible** $(d, 3)$-graphlets, we use Comb_Three which takes $O(|V|)$ time (Lemma 5).

---

**Algorithm 6:** GET_TRIANGLE

**Input:** (1) Three nodes consisting a triangle $u, v, w$
         (2) Maximum distance considered $d$
         (3) $d$-DAG $\vec{G}^{(d)} = (V, \vec{E}^{(\leq d)})$ of graph $G$

**Output:** The corresponding $(d, s)$-graphlet $T_*^{(d)} \in T^{(d)}$

    // Retrieve pairwise distances (Algorithm 4)

1   $d_i \leftarrow$ RETRIEVE_DISTANCE$((u, v), d, \vec{G}^{(d)})$

2   $d_j \leftarrow$ RETRIEVE_DISTANCE$((u, w), d, \vec{G}^{(d)})$

3   $d_k \leftarrow$ RETRIEVE_DISTANCE$((v, w), d, \vec{G}^{(d)})$

    // $(2, 3)$-graphlets $(d = 2)$

4   **if** $d = 2$

5      **if** $(d_i, d_j, d_k) \in \{(2, 2, 2)\}$

6          $T_*^{(2)} \leftarrow T_1^{(2)}$

7      **else if** $(d_i, d_j, d_k) \in \{(1, 2, 2), (2, 1, 2), (2, 2, 1)\}$

8          $T_*^{(2)} \leftarrow T_2^{(2)}$

9      **else if** $(d_i, d_j, d_k) \in \{(2, 1, 1), (1, 2, 1), (1, 1, 2)\}$

10         $T_*^{(2)} \leftarrow T_3^{(2)}$

11      **else if** $(d_i, d_j, d_k) \in \{(1, 1, 1)\}$

12         $T_*^{(2)} \leftarrow T_4^{(2)}$

    // $(3, 3)$-graphlets $(d = 3)$

13   **if** $d = 3$

14      **if** $(d_i, d_j, d_k) \in \{(3, 3, 3)\}$

15         $T_*^{(3)} \leftarrow T_1^{(3)}$

16      **else if** $(d_i, d_j, d_k) \in \{(2, 3, 3), (3, 2, 3), (3, 3, 2)\}$

17         $T_*^{(3)} \leftarrow T_2^{(3)}$

18      **else if** $(d_i, d_j, d_k) \in \{(2, 2, 3), (2, 3, 2), (3, 2, 2)\}$

19         $T_*^{(3)} \leftarrow T_3^{(3)}$

20      **else if** $(d_i, d_j, d_k) \in \{(1, 3, 3), (3, 1, 3), (3, 3, 1)\}$

21         $T_*^{(3)} \leftarrow T_4^{(3)}$

22      **else if** $(d_i, d_j, d_k) \in \{(2, 2, 2)\}$

23         $T_*^{(3)} \leftarrow T_6^{(3)}$

24      **else if** $(d_i, d_j, d_k) \in \{(1, 2, 2), (2, 1, 2), (2, 2, 1)\}$

25         $T_*^{(3)} \leftarrow T_7^{(3)}$

26      **else if** $(d_i, d_j, d_k) \in \{(1, 1, 1)\}$

27         $T_*^{(3)} \leftarrow T_9^{(3)}$

28   **return** $T_*^{(d)}$

---

Since counting non-deducible $(d, s)$-graphlets dominates the entire complexity, the total time complexity of Algorithm 1 is $O(|V|d^4\vec{\Delta}^{2d} \log \vec{\Delta})$.

## A.4   Details of Algorithm 2

In this subsection, we provide the details of sub-algorithms used in Algorithm 2. Then, we provide the proof of Theorem 2 regarding the complexity of Algorithm 2.

**TRIANGLE_PAIRS (Algorithm 8).** This function identifies the set of *effective* triangle pairs formed by the edge $(u, v)$, where the remaining nodes are connected. Specifically, we consider a pair of triangles $(u, v, w)$ and $(u, v, w')$ effective if $w$ and $w'$ for a 1-edge or 2-edge. The time complexity of Algorithm 8 is provided in Lemma 6.

**LEMMA 6 (COMPLEXITY OF ALGORITHM 8).** *The time complexity of retrieving effective triangle pairs is $O(\vec{\Delta}^4 \log \vec{\Delta})$, where $\vec{\Delta}$ is the maximum out-degree, i.e., $\vec{\Delta} = \max_{u \in V} |\vec{N}_u^{(1)}|$.*

**PROOF.** It retrieves the distance for every common neighboring pair $(w, w')$ of $(u, v)$, which takes $O(|N_{u, v}|^2) = O(\vec{\Delta}^{2d})$, where $\vec{\Delta}$ is the maximum out-degree, i.e., $\vec{\Delta} = \max_{u \in V} |\vec{N}_u^{(1)}|$. For each pair, retrieving the distance between $w$ and $w'$ takes $O(d^2 \log \vec{\Delta})$ time (Lemma 2). Thus, the total time complexity is $O(\vec{\Delta}^{2d} d^2 \log \vec{\Delta}) = O(\vec{\Delta}^4 \log \vec{\Delta})$ since we assume $d = 2$.

**GET_CLIQUE (Algorithm 9).** Given two triangles ($(2, 3)$-graphlets) $T_\circ^{(2)}, T_\bullet^{(2)}$ and four nodes $(u, v, w, w')$ that form a clique, this function returns the corresponding $(2, 4)$-graphlet of the clique. It first retrieves the distances of the additional necessary edge pairs. Then, based on these four distances and the $(2, 3)$-graphlets of the two triangles, it immediately identifies the $(2, 4)$-graphlet of the clique. The time complexity of Algorithm 9 is provided in Lemma 7.

**LEMMA 7 (COMPLEXITY OF ALGORITHM 9).** *The time complexity of identifying $(2, 4)$-graphlet of a clique is $O(\log \vec{\Delta})$, where $\vec{\Delta}$ is the maximum out-degree, i.e., $\vec{\Delta} = \max_{u \in V} |\vec{N}_u^{(1)}|$.*

**PROOF.** The time complexity for retrieving the distances of pairs of nodes in the clique is $O(d^2 \log \vec{\Delta})$ (from Lemma 2). Once the distances are obtained, the corresponding $(2, 4)$-graphlet can be identified in $O(1)$ time. Thus, the overall time complexity is $O(\log \vec{\Delta})$ assuming that we use $d = 2$.

**GET_NON-INDUCED_WEDGE (Algorithm 11).** This function identifies the type of non-induced wedges (which can either be a wedge or a triangle) for a given set of three nodes $(u, v, w)$. Here, we assume that $(u, w)$ is disconnected, and focus on identifying the wedge (i.e., $T_5^{(5)}$ and $T_5^{(6)}$) formed by the triple of nodes. To this end, we retrieve the distances between $(u, v)$ and $(v, w)$ and then rapidly identify the corresponding wedge based on these distances. The time complexity of Algorithm 11 is provided in Lemma 8.

**LEMMA 8 (COMPLEXITY OF ALGORITHM 11).** *The time complexity of identifying $T_5^{(2)}$ and $T_6^{(2)}$ of the given triple of nodes $(u, v, w)$, assuming that $u$ and $w$ are disconnected, is $O(\log \vec{\Delta})$, where $\vec{\Delta}$ is the maximum out-degree, i.e., $\vec{\Delta} = \max_{u \in V} |\vec{N}_u^{(1)}|$.*

**PROOF.** The time complexity for retrieving the distances of pairs of nodes in the clique is $O(d^2 \log \vec{\Delta})$ (from Lemma 2). Once the distances are obtained, the corresponding wedge ($T_5^{(2)}$ and $T_6^{(2)}$) can be identified in $O(1)$ time. Thus, the overall time complexity is $O(\log \vec{\Delta})$ assuming that we use $d = 2$.

**GET_NON-INDUCED_CYCLE (Algorithm 12).** A non-induced cycle can be obtained based on the predefined conditions of the given pair of wedges. Thus, the time complexity of Algorithm 12 is $O(1)$.

**COMB_FOUR (Algorithm 13).** This function computes the counts of deducible $(2, 4)$-graphlets ($\widehat{Q}^{(2)}$) and adjusts the counts of semi-deducible $(2, 4)$-graphlets ($\overline{Q}^{(2)}$). Specifically, it leverages node degrees or edge counts to quickly compute these values. In the worst case, enumeration over the $E^{(2)}$. The time complexity of Algorithm 13 is provided in Lemma 9.

**Algorithm 7:** COMB_THREE

**Input:** (1) Target deducible $(d, s)$-graphlet $T_j^{(d)} \in \widehat{T}^{(d)}$
     (2) Intermediate counts of $(d, s)$-graphlets $\{C(T_i^{(d)})\}_{i=1}^{|T^{(d)}|}$
     (3) $d$-graph $G^{(d)} = (V, E^{(\leq d)})$ of graph $G$

**Output:** The count of the target $(d, s)$-graphlet $T_j^{(d)}$

// (2,3)-graphlets ($d = 2$)

1 **if** $G^{(d)} = G^{(2)}$

    // Apply the appropriate equation to $T_j^{(2)}$. The equations should be applied in the below following order.

2    $C(T_3^{(2)}) \leftarrow \sum_{u \in V} \binom{|N_u^{(1)}|}{2} - 3C(T_4^{(2)})$

3    $C(T_5^{(2)}) \leftarrow \sum_{u \in V} \binom{|N_u^{(2)}|}{2} - 3C(T_1^{(2)}) - C(T_2^{(2)})$

4    $C(T_6^{(2)}) \leftarrow \sum_{u \in V} (|N_u^{(1)}| \, |N_u^{(2)}|) - 2C(T_2^{(2)}) - 2C(T_3^{(2)})$

// (3,3)-graphlets ($d = 3$)

5 **if** $G^{(d)} = G^{(3)}$

    // Apply the appropriate equation to $T_j^{(3)}$. The equations should be applied in the below following order.

6    $C(T_8^{(3)}) \leftarrow \sum_{u \in V} \binom{|N_u^{(1)}|}{2} - 3C(T_9^{(3)})$

7    $C(T_5^{(3)}) \leftarrow \sum_{u \in V} (|N_u^{(1)}| \, |N_u^{(2)}|) - 2C(T_7^{(3)}) - 2C(T_8^{(3)})$

8    $C(T_{10}^{(3)}) \leftarrow \sum_{u \in V} \binom{|N_u^{(3)}|}{2} - 3C(T_1^{(3)}) - C(T_2^{(3)}) - C(T_4^{(3)})$

9    $C(T_{11}^{(3)}) \leftarrow \sum_{u \in V} (|N_u^{(2)}| \, |N_u^{(3)}|) - 2C(T_2^{(3)}) - 2C(T_3^{(3)}) - 2C(T_5^{(3)})$

10    $C(T_{12}^{(3)}) \leftarrow \sum_{u \in V} \binom{|N_u^{(2)}|}{2} - C(T_3^{(3)}) - 3C(T_6^{(3)}) - C(T_7^{(3)})$

11    $C(T_{13}^{(3)}) \leftarrow \sum_{u \in V} (|N_u^{(1)}| \, |N_u^{(3)}|) - 2C(T_4^{(3)}) - C(T_5^{(3)})$

---

**Algorithm 8:** TRIANGLE_PAIRS

**Input:** (1) Two nodes consisting an edge $(u, v)$
     (2) Common neighbor nodes set between $u$ and $v : N_{u,v}$
     (3) 2-DAG $\vec{G}^{(2)} = (V, \vec{E}^{(\leq 2)})$ of graph $G$

**Output:** The set of *effective* pairs triangles $\{(u, v, w), (u, v, w')\}$
     that share nodes $u$ and $v : \mathcal{T}_{u,v}$

1 $\mathcal{T}_{u,v} \leftarrow \emptyset$

2 **for each** $(w, w') \in \binom{N_{u,v}}{2}$

3    $\delta(w, w') \leftarrow$ RETRIEVE_DISTANCE$((w, w'), 2, \vec{G}^{(2)})$

4    **if** $\delta(w, w') \neq \infty$

5        $\mathcal{T}_{u,v} \leftarrow \mathcal{T}_{u,v} \cup \{\{(u, v, w), (u, v, w')\}\}$

6 **return** $\mathcal{T}_{u,v}$

---

**LEMMA 9 (COMPLEXITY OF ALGORITHM 13).** *The time complexity of computing the counts of deducible $(2, 4)$-graphlets and adjusting the counts of semi-deducible $(2, 4)$-graphlet is $O(|V|\Delta^2)$.*

**PROOF.** In the worst case, it requires enumeration over $E^{(2)}$, and thus the time complexity is $O(|E|^{(2)}) = O(|V|\Delta^2)$.

**Proof of Theorem 2.** Below, we provide the proof for Theorem 2.

**PROOF.** The time complexity of Algorithm 2 is determined by three main operations: (1) computing non-deducible $(2, 4)$-graphlets, (2) computing semi-deducible $(2, 4)$-graphlets, and (3) computing deducible $(2, 4)$-graphlets.

- To count **non-deducible** $(2, 4)$-graphlets, we iterate over each node $u \in V$. For each of $u$'s neighbor $v \in \vec{N}_u^{(1)} \cup \vec{N}_u^{(2)}$, we first compute the common neighbors $N_{u,v}$ which takes $O(\min(|\vec{N}_u^{(1)} \cup \vec{N}_u^{(2)}|, |\vec{N}_v^{(1)} \cup \vec{N}_v^{(2)}|)) = O(\vec{\Delta}^2)$ time. Using the common neighbors, the effective triangle pairs are retrieved using TRIANGLE_PAIRS, which takes $O(\vec{\Delta}^4 \log \vec{\Delta})$ time (Lemma 6), and the number of pairs is $O(\vec{\Delta}^4)$. Then for each triangle pair, each of the corresponding triangle's $(2, 3)$-graphlet is identified. Using these $(2, 3)$-graphlets, the clique is then determined, which takes $O(\log \vec{\Delta})$ time (Lemma 7). Thus, the time complexity of counting non-deducible $(2, 4)$-graphlets is $(|V|\vec{\Delta}^4 \log \vec{\Delta})$.

- To count **semi-deducible** $(2, 4)$-graphlets, for each node $v$, we enumerate over its neighboring pairs $(v, v')$, which takes $O(\vec{\Delta}^4)$ time. Then we iterate each of the $O(\Delta^2)$ common neighbors $w$ of $v$ and $v'$, and identify the non-induced wedge which takes $O(\log \vec{\Delta})$ time (Lemma 8). Then the cycle is identified in $O(1)$ time. Thus, the time complexity of counting semi-deducible $(2, 4)$-graphlets is $(|V|\vec{\Delta}^4\Delta^2 \log \vec{\Delta})$.

- To count **deducible** $(2, 4)$-graphlets, we use the COMB_FOUR which takes $O(|V|\vec{\Delta}^2)$ time.

As a result, counting semi-deducible $(d, s)$-graphlets dominate the entire complexity, and thus the overall time complexity of Algorithm 2 is $(|V|\vec{\Delta}^4\Delta^2 \log \vec{\Delta})$.

## B EXPERIMENT DETAILS

In this section, we provide further details on our experiments.

**Algorithm 9:** GET_CLIQUE

**Input:** (1) Two triangle types $T_\circ^{(2)}, T_\bullet^{(2)}$
   (2) Four nodes consisting a clique $u, v, w, w'$
   (3) 2-DAG $\vec{G}^{(2)} = (V, \vec{E}^{(\leq 2)})$ of graph $G$
**Output:** The corresponding $(d, s)$-graphlet $Q_*^{(d)} \in \overline{Q}^{(d)}$

1   $d_{(u,v)} \leftarrow$ RETRIEVE_DISTANCE$(u, v, 2, \vec{G}^{(2)})$
2   $d_{(u,w)} \leftarrow$ RETRIEVE_DISTANCE$(u, w, 2, \vec{G}^{(2)})$
3   $d_{(u,w')} \leftarrow$ RETRIEVE_DISTANCE$(u, w', 2, \vec{G}^{(2)})$
4   $d_{(w,w')} \leftarrow$ RETRIEVE_DISTANCE$(w, w', 2, \vec{G}^{(2)})$
5   **if** $d_{(u,v)} = 2$
6    **if** $(T_\circ^{(2)}, T_\bullet^{(2)}, d_{(w,w')}) = (T_1^{(2)}, T_1^{(2)}, 2)$
7    $\lfloor\ Q_*^{(2)} \leftarrow Q_1^{(2)}$
8    **else if** $(T_\circ^{(2)}, T_\bullet^{(2)}, d_{(w,w')}) = (T_1^{(2)}, T_1^{(2)}, 1)$
9    $\lfloor\ Q_*^{(2)} \leftarrow Q_2^{(2)}$
10    **else if** $(T_\circ^{(2)}, T_\bullet^{(2)}, d_{(w,w')}) = (T_1^{(2)}, T_2^{(2)}, 2)$
11    $\lfloor\ Q_*^{(2)} \leftarrow Q_2^{(2)}$
12    **else if** $(T_\circ^{(2)}, T_\bullet^{(2)}, d_{(w,w')}) = (T_1^{(2)}, T_2^{(2)}, 1)$
13    $\lfloor\ Q_*^{(2)} \leftarrow Q_4^{(2)}$
14    **else if** $(T_\circ^{(2)}, T_\bullet^{(2)}, d_{(w,w')}) = (T_1^{(2)}, T_3^{(2)}, 2)$
15    $\lfloor\ Q_*^{(2)} \leftarrow Q_4^{(2)}$
16    **else if** $(T_\circ^{(2)}, T_\bullet^{(2)}, d_{(w,w')}) = (T_1^{(2)}, T_3^{(2)}, 1)$
17    $\lfloor\ Q_*^{(2)} \leftarrow Q_6^{(2)}$
18    **else if** $(T_\circ^{(2)}, T_\bullet^{(2)}, d_{(w,w')}) = (T_2^{(2)}, T_2^{(2)}, 2)$
19    **if** $d_{(u,w)} = d_{(u,w')}$
20     $\lfloor\ Q_*^{(2)} \leftarrow Q_4^{(2)}$
21    **else**
22     $\lfloor\ Q_*^{(2)} \leftarrow Q_3^{(2)}$
23    **else if** $(T_\circ^{(2)}, T_\bullet^{(2)}, d_{(w,w')}) = (T_2^{(2)}, T_2^{(2)}, 1)$
24    **if** $d_{(u,w)} = d_{(u,w')}$
25     $\lfloor\ Q_*^{(2)} \leftarrow Q_7^{(2)}$
26    **else**
27     $\lfloor\ Q_*^{(2)} \leftarrow Q_5^{(2)}$
28    **else if** $(T_\circ^{(2)}, T_\bullet^{(2)}, d_{(w,w')}) = (T_2^{(2)}, T_3^{(2)}, 2)$
29    $\lfloor\ Q_*^{(2)} \leftarrow Q_5^{(2)}$
30    **else if** $(T_\circ^{(2)}, T_\bullet^{(2)}, d_{(w,w')}) = (T_2^{(2)}, T_3^{(2)}, 1)$
31    $\lfloor\ Q_*^{(2)} \leftarrow Q_9^{(2)}$
32    **else if** $(T_\circ^{(2)}, T_\bullet^{(2)}, d_{(w,w')}) = (T_3^{(2)}, T_3^{(2)}, 2)$
33    $\lfloor\ Q_*^{(2)} \leftarrow Q_8^{(2)}$
34    **else if** $(T_\circ^{(2)}, T_\bullet^{(2)}, d_{(w,w')}) = (T_3^{(2)}, T_3^{(2)}, 1)$
35    $\lfloor\ Q_*^{(2)} \leftarrow Q_{10}^{(2)}$
   `// Continue next page`

## B.1 Datasets

The details of the datasets and domains are provided below:

- collaboration (ca-DBLP [13], ca-Citeseer [3, 25], ca-HepTh [34]):
  Collaboration networks from various academic fields, where
  nodes represent authors and edges represent co-authorship be-
  tween two authors.

**Algorithm 10:** GET_CLIQUE (CONTINUED)

1   **else**
2   **if** $(T_\circ^{(2)}, T_\bullet^{(2)}, d_{(w,w')}) = (T_2^{(2)}, T_2^{(2)}, 2)$
3    $\lfloor\ Q_*^{(2)} \leftarrow Q_2^{(2)}$
4   **else if** $(T_\circ^{(2)}, T_\bullet^{(2)}, d_{(w,w')}) = (T_2^{(2)}, T_2^{(2)}, 1)$
5    $\lfloor\ Q_*^{(2)} \leftarrow Q_3^{(2)}$
6   **else if** $(T_\circ^{(2)}, T_\bullet^{(2)}, d_{(w,w')}) = (T_2^{(2)}, T_3^{(2)}, 2)$
7    $\lfloor\ Q_*^{(2)} \leftarrow Q_4^{(2)}$
8   **else if** $(T_\circ^{(2)}, T_\bullet^{(2)}, d_{(w,w')}) = (T_2^{(2)}, T_3^{(2)}, 1)$
9    $\lfloor\ Q_*^{(2)} \leftarrow Q_5^{(2)}$
10   **else if** $(T_\circ^{(2)}, T_\bullet^{(2)}, d_{(w,w')}) = (T_2^{(2)}, T_4^{(2)}, 2)$
11    $\lfloor\ Q_*^{(2)} \leftarrow Q_7^{(2)}$
12   **else if** $(T_\circ^{(2)}, T_\bullet^{(2)}, d_{(w,w')}) = (T_2^{(2)}, T_4^{(2)}, 1)$
13    $\lfloor\ Q_*^{(2)} \leftarrow Q_9^{(2)}$
14   **else if** $(T_\circ^{(2)}, T_\bullet^{(2)}, d_{(w,w')}) = (T_3^{(2)}, T_3^{(2)}, 2)$
15    **if** $d_{(u,w)} = d_{(u,w')}$
16    $\lfloor\ Q_*^{(2)} \leftarrow Q_6^{(2)}$
17    **else**
18    $\lfloor\ Q_*^{(2)} \leftarrow Q_5^{(2)}$
19   **else if** $(T_\circ^{(2)}, T_\bullet^{(2)}, d_{(w,w')}) = (T_3^{(2)}, T_3^{(2)}, 1)$
20    **if** $d_{(u,w)} = d_{(u,w')}$
21    $\lfloor\ Q_*^{(2)} \leftarrow Q_9^{(2)}$
22    **else**
23    $\lfloor\ Q_*^{(2)} \leftarrow Q_8^{(2)}$
24   **else if** $(T_\circ^{(2)}, T_\bullet^{(2)}, d_{(w,w')}) = (T_3^{(2)}, T_4^{(2)}, 2)$
25    $\lfloor\ Q_*^{(2)} \leftarrow Q_9^{(2)}$
26   **else if** $(T_\circ^{(2)}, T_\bullet^{(2)}, d_{(w,w')}) = (T_3^{(2)}, T_4^{(2)}, 1)$
27    $\lfloor\ Q_*^{(2)} \leftarrow Q_{10}^{(2)}$
28   **else if** $(T_\circ^{(2)}, T_\bullet^{(2)}, d_{(w,w')}) = (T_4^{(2)}, T_4^{(2)}, 2)$
29    $\lfloor\ Q_*^{(2)} \leftarrow Q_{10}^{(2)}$
30   **else if** $(T_\circ^{(2)}, T_\bullet^{(2)}, d_{(w,w')}) = (T_4^{(2)}, T_4^{(2)}, 1)$
31    $\lfloor\ Q_*^{(2)} \leftarrow Q_{11}^{(2)}$
32   **return** $Q_*^{(2)}$

**Algorithm 11:** GET_NON-INDUCED_WEDGE

**Input:** (1) Three nodes consisting a wedge $(u, v, w)$
   (2) $d$-DAG $\vec{G}^{(d)} = (V, \vec{E}^{(\leq d)})$ of graph $G$
**Output:** The corresponding $(d, s)$-graphlet $T_*^{(2)} \in T^{(2)}$

1   $\delta(u, v) \leftarrow$ RETRIEVE_DISTANCE$(u, v, 2, \vec{G}^{(2)})$
2   $\delta(v, w) \leftarrow$ RETRIEVE_DISTANCE$(v, w, 2, \vec{G}^{(2)})$
3   **if** $(\delta(u, v), \delta(v, w)) \in \{(1, 2), (2, 1)\}$
4   $\lfloor\ T_*^{(2)} \leftarrow T_6^{(2)}$
5   **else**
6   $\lfloor\ T_*^{(2)} \leftarrow T_5^{(2)}$
7   **return** $T_*^{(2)}$

- web [11, 12, 54] (web-Arabic, web-Indochina): Web networks,
  where nodes represent web pages and edges represent hyperlinks
  between pages.

**Algorithm 12:** GET_NON-INDUCED_CYCLE

**Input:** (1) Two wedge type $T_\triangle^{(2)}, T_\blacktriangle^{(2)}$
**Output:** The corresponding $(d, s)$-graphlet $Q_*^{(2)} \in \widetilde{Q}^{(2)}$

1 **if** $(T_\triangle^{(2)}, T_\blacktriangle^{(2)}) \in \{(T_5^{(2)}, T_5^{(2)})\}$
2     $Q_*^{(2)} \leftarrow Q_{30}^{(2)}$
3 **else if** $(T_\triangle^{(2)}, T_\blacktriangle^{(2)}) \in \{(T_5^{(2)}, T_6^{(2)}), (T_6^{(2)}, T_5^{(2)})\}$
4     $Q_*^{(2)} \leftarrow Q_{29}^{(2)}$
5 **else**
6     $Q_*^{(2)} \leftarrow Q_{28}^{(2)}$
7 **return** $Q_*^{(2)}$

**Table 4: Statistics for 13 real-world graphs across 5 domains: $|E^{(d)}|$ is the number of $d$-edges, and $|T^{(d)}|$ and $|Q^{(d)}|$ are the counts of size-3 and size-4 $(d, s)$-graphlets, respectively.**

| Dataset | $|V|$ | $|E^{(1)}|$ | $|E^{(2)}|$ | $|E^{(3)}|$ | $|T^{(2)}|$ | $|T^{(3)}|$ | $|Q^{(2)}|$ |
|---|---|---|---|---|---|---|---|
| ca-DBLP | 317K | 1.05M | 12.7M | 153M | 4.89B | 769B | 3.95T |
| ca-Citeseer | 227K | 814K | 7.38M | 50.4M | 1.70B | 107B | 659B |
| ca-HepTh | 9.88K | 26.0K | 179K | 1.10M | 21.8M | 797M | 3.77B |
| web-Arabic | 164K | 1.75M | 3.06M | 14.9M | 755M | 12.6B | 205B |
| web-Indochina | 11.4K | 47.6K | 425K | 3.70M | 121M | 2.93B | 54.5B |
| soc-UCSC | 8.99K | 225K | 7.19M | 24.8M | 12.7B | 98.9B | 21.9T |
| soc-UC | 6.83K | 155K | 4.72M | 13.5M | 7.13B | 42.5B | 9.80T |
| soc-MB | 3.08K | 125K | 2.35M | 1.96M | 2.34B | 4.62B | 1.79T |
| tags-Ubuntu | 3.03K | 133K | 3.66M | 764K | 3.96B | 4.59B | 3.07T |
| tags-Math | 1.63K | 91.7K | 1.08M | 152K | 661M | 716M | 275B |
| road-CA | 1.97M | 2.77M | 5.12M | 8.07M | 45.0M | 189M | 301M |
| road-PA | 1.09M | 1.54M | 2.88M | 4.58M | 25.7M | 109M | 175M |
| road-TX | 1.38M | 1.92M | 3.52M | 5.55M | 30.7M | 128M | 202M |

**Table 5: Importance scores of $(2, 4)$-graphlets for each dataset. Each rank is based on the importance score, and each value represents the index of graphlet instance, with the score shown in parentheses.**

| Dataset | 1st | 2nd | 3rd | 4th | 5th |
|---|---|---|---|---|---|
| ca-DBLP | 4 (0.91) | 13 (0.90) | 2 (0.86) | 3 (0.85) | 27 (0.76) |
| ca-Citeseer | 4 (0.82) | 13 (0.81) | 3 (0.76) | 5 (0.76) | 2 (0.70) |
| ca-HepTh | 4 (0.89) | 13 (0.89) | 2 (0.84) | 5 (0.79) | 3 (0.79) |
| web-Arabic | 3 (0.99) | 5 (0.98) | 36 (0.94) | 12 (0.94) | 8 (0.93) |
| web-Indochina | 3 (0.99) | 5 (0.98) | 36 (0.94) | 12 (0.94) | 8 (0.94) |
| soc-UCSC | 5 (0.96) | 4 (0.91) | 3 (0.89) | 2 (0.89) | 1 (0.87) |
| soc-UC | 5 (0.95) | 4 (0.93) | 3 (0.91) | 2 (0.91) | 6 (0.89) |
| soc-MB | 5 (0.93) | 27 (0.89) | 18 (0.88) | 4 (0.85) | 6 (0.84) |
| tags-Ubuntu | 6 (0.96) | 20 (0.94) | 3 (0.94) | 22 (0.93) | 28 (0.91) |
| tags-Math | 6 (0.96) | 3 (0.94) | 20 (0.93) | 22 (0.92) | 14 (0.91) |
| road-CA | 30 (0.99) | 17 (0.99) | 29 (0.99) | 12 (0.99) | 28 (0.99) |
| road-PA | 17 (0.99) | 30 (0.99) | 12 (0.99) | 14 (0.99) | 29 (0.99) |
| road-TX | 30 (0.99) | 29 (0.99) | 17 (0.99) | 28 (0.99) | 12 (0.99) |

- social-Facebook [59, 60] (soc-UCSC, soc-UC, soc-MB): Social friendship networks from Facebook at various US schools, where

nodes represent users and edges represent friendship connections between them.
- tags [7] (tags-Ubuntu, tags-Math): Tag co-occurrence networks from question-and-answer sites, where nodes represent tags and edges link tags that appear together on the same question post.
- road [36] (road-CA, road-PA, road-TX): Road networks from various US regions, where nodes represent intersections or road endpoints, and edges represent the roads connecting them.

We removed self-loops for our analysis. The preprocessed datasets can be accessed at [26]. All original datasets used in this study are publicly available from [7, 35, 54]. We present the dataset statistics, including the number of nodes, edges, and graphlet instances, in Table 4.

## B.2 Importance scores

Among the $(2, 4)$-graphlets, $Q_6^{(2)}, Q_8^{(2)}, Q_9^{(2)}, Q_{10}^{(2)}, Q_{11}^{(2)}, Q_{19}^{(2)}$ can also be represented in the original size-4 graphlet. Since the 2-edges of these instances can be inferred from the 1-edge (i.e., the distance between nodes that are not connected by a 1-edge is guaranteed to be at most 2.), they are naturally represented in the $(2, 4)$-graphlets as well. The remaining $(2, 4)$-graphlet instances are newly captured local structures, identified by considering distances up to 2. To understand how the newly defined $(d, s)$-graphlet plays a significant role in characterization, we use the scoring function proposed by [32], which denotes the importance of each graphlet $g$.

$$Importance(g) = 1 - \frac{dist_{within}(g)}{dist_{across}(g)}$$

$dist_{within}(g)$ is the average CP distance between other graphs from the same domain, and $dist_{across}(g)$ is the average CP distance between other graphs from different domains. We calculate the importance of $(2, 4)$-graphlets across all 13 datasets and display the top 5 instances for each dataset in Table 5, where the graphlet indices are ranked by the importance score.

## B.3 Exact Counting time for all algorithms

In Section 6.4, we evaluate the counting time of EDGE from two perspectives: (1) in comparison to conventional graphlet counting algorithms (PGD and ESCAPE) for size-4 $(d, s)$-graphlets, and (2) against two ablation variants of EDGE (EDGE-D2 and EDGE-D). Table 6 presents the exact counting time of all algorithms along with additional information about graphlet instances.

---

**Algorithm 13:** Comb_Four

**Input:** (1) Target deducible or semi-deducible $(2,4)$-graphlet $Q_j^{(2)} \in \widehat{Q}^{(2)} \cup \widetilde{Q}^{(2)}$

(2) Intermediate counts of $(d,s)$-graphlets $\{C(Q_i^{(d)})\}_{i=1}^{|Q^{(d)}|}$

(3) The counts of each $T_i^{(2)}$ per edge $\{C_e(T_i^{(2)})\}_{i=1}^{|T^{(2)}|}$

(4) 2-graph $G^{(2)} = (V, E^{(\leq 2)})$ of graph $G$

**Output:** The count of the target $(d,s)$-graphlet $Q_j^{(d)}$

// Apply the appropriate equation to $Q_j^{(2)}$. The equations should be applied in the below following order.

1   $C(Q_{12}^{(2)}) \leftarrow \sum_{(u,v) \in E^{(2)}} \binom{C_{(u,v)}(T_1^{(2)})}{2} - 6C(Q_1^{(2)}) - C(Q_2^{(2)})$

2   $C(Q_{13}^{(2)}) \leftarrow \sum_{(u,v) \in E^{(1)}} \binom{C_{(u,v)}(T_2^{(2)})}{2} - C(Q_2^{(2)}) - 2C(Q_3^{(2)})$

3   $C(Q_{14}^{(2)}) \leftarrow \sum_{(u,v) \in E^{(2)}} (C_{(u,v)}(T_1^{(2)}) C_{(u,v)}(T_2^{(2)})) - 4C(Q_2^{(2)}) - 2C(Q_4^{(2)})$

4   $C(Q_{15}^{(2)}) \leftarrow \sum_{(u,v) \in E^{(2)}} \binom{C_{(u,v)}(T_2^{(2)})}{2} - 4C(Q_3^{(2)}) - C(Q_5^{(2)}) - C(Q_4^{(2)}) - 3C(Q_7^{(2)})$

5   $C(Q_{16}^{(2)}) \leftarrow \sum_{(u,v) \in E^{(1)}} (C_{(u,v)}(T_2^{(2)}) C_{(u,v)}(T_3^{(2)})) - 2C(Q_4^{(2)}) - 2C(Q_5^{(2)})$

6   $C(Q_{17}^{(2)}) \leftarrow \sum_{(u,v) \in E^{(1)}} (C_{(u,v)}(T_1^{(2)}) C_{(u,v)}(T_3^{(2)})) - C(Q_4^{(2)}) - 3C(Q_6^{(2)})$

7   $C(Q_{18}^{(2)}) \leftarrow \sum_{(u,v) \in E^{(1)}} (C_{(u,v)}(T_2^{(2)}) C_{(u,v)}(T_4^{(2)})) - 3C(Q_7^{(2)}) - C(Q_9^{(2)})$

8   $C(Q_{19}^{(2)}) \leftarrow \sum_{(u,v) \in E^{(1)}} \binom{C_{(u,v)}(T_3^{(2)})}{2} - C(Q_5^{(2)}) - 4C(Q_8^{(2)}) - 3C(Q_6^{(2)}) - C(Q_9^{(2)})$

9   $C(Q_{20}^{(2)}) \leftarrow \sum_{(u,v) \in E^{(2)}} \{C_{(u,v)}(T_1^{(2)})(|N_u^{(2)}| + |N_v^{(2)}|)\} - 4C(Q_{12}^{(2)}) - C(Q_{14}^{(2)}) - 12C(Q_1^{(2)}) - 4C(Q_2^{(2)}) - C(Q_4^{(2)})$

10   $C(Q_{21}^{(2)}) \leftarrow \sum_{(u,v) \in E^{(2)}} \{C_{(u,v)}(T_1^{(2)})(|N_u^{(1)}| + |N_v^{(1)}|)\} - C(Q_{14}^{(2)}) - 2C(Q_{17}^{(2)}) - 2C(Q_2^{(2)}) - 2C(Q_4^{(2)}) - 3C(Q_6^{(2)})$

11   $C(Q_{22}^{(2)}) \leftarrow \sum_{(u,v) \in E^{(1)}} \{C_{(u,v)}(T_2^{(2)})(|N_u^{(2)}| + |N_v^{(2)}|)\} - 4C(Q_{13}^{(2)}) - C(Q_{14}^{(2)}) - 2C(Q_{15}^{(2)}) - C(Q_{16}^{(2)}) - 4C(Q_2^{(2)}) - 8C(Q_3^{(2)}) - 2C(Q_4^{(2)}) - 2C(Q_5^{(2)})$

12   $C(Q_{23}^{(2)}) \leftarrow \sum_{(u,v) \in E^{(2)}} \{C_{(u,v)}(T_2^{(2)})(|N_u^{(2)}| + |N_v^{(2)}|)\} - \sum_{(u,v) \in E^{(1)}} \{C_{(u,v)}(T_2^{(2)})(|N_u^{(2)}| + |N_v^{(2)}|)\} - C(Q_{14}^{(2)}) - 2C(Q_2^{(2)}) - 2C(Q_4^{(2)}) - 3C(Q_7^{(2)})$

13   $C(Q_{24}^{(2)}) \leftarrow \sum_{(u,v) \in E^{(2)}} \{C_{(u,v)}(T_2^{(2)})(|N_u^{(1)}| + |N_v^{(1)}|)\} - \sum_{(u,v) \in E^{(1)}} \{C_{(u,v)}(T_2^{(2)})(|N_u^{(1)}| + |N_v^{(1)}|)\} - 2C(Q_{15}^{(2)}) - 4C(Q_3^{(2)}) - 2C(Q_5^{(2)}) - C(Q_9^{(2)})$

14   $C(Q_{25}^{(2)}) \leftarrow \sum_{(u,v) \in E^{(1)}} \{C_{(u,v)}(T_3^{(2)})(|N_u^{(2)}| + |N_v^{(2)}|)\} - \sum_{(u,v) \in E^{(2)}} \{C_{(u,v)}(T_3^{(2)})(|N_u^{(2)}| + |N_v^{(2)}|)\} - C(Q_{16}^{(2)}) - 2C(Q_{19}^{(2)})$

15       $-C(Q_4^{(2)}) - 2C(Q_5^{(2)}) - 4C(Q_8^{(2)})$

16   $C(Q_{26}^{(2)}) \leftarrow \sum_{(u,v) \in E^{(2)}} \{C_{(u,v)}(T_3^{(2)})(|N_u^{(2)}| + |N_v^{(2)}|)\} - C(Q_{16}^{(2)}) - 2 \cdot C(Q_{17}^{(2)}) - 2 \cdot C(Q_4^{(2)}) - 2 \cdot C(Q_5^{(2)}) - 6 \cdot C(Q_6^{(2)}) - 2 \cdot C(Q_9^{(2)})$

17   $C(Q_{27}^{(2)}) \leftarrow \sum_{(u,v) \in E^{(1)}} \{C_{(u,v)}(T_4^{(2)})(|N_u^{(2)}| + |N_v^{(2)}|)\} - 2 \cdot C(Q_{18}^{(2)}) - C(Q_9^{(2)}) - 2 \cdot C(Q_{10}^{(2)}) - 3 \cdot C(Q_7^{(2)}) - C(Q_9^{(2)})$

18   $C(Q_{28}^{(2)}) \leftarrow C(Q_{28}^{(2)}) - C(Q_{12}^{(2)}) - C(Q_{13}^{(2)}) - 3 \cdot C(Q_1^{(2)}) - C(Q_2^{(2)}) - C(Q_3^{(2)})$

19   $C(Q_{29}^{(2)}) \leftarrow C(Q_{29}^{(2)}) - C(Q_{14}^{(2)}) - C(Q_{16}^{(2)}) - 2 \cdot C(Q_2^{(2)}) - 2 \cdot C(Q_4^{(2)}) - C(Q_5^{(2)})$

20   $C(Q_{30}^{(2)}) \leftarrow C(Q_{30}^{(2)}) - C(Q_{15}^{(2)}) - C(Q_{19}^{(2)}) - 2C(Q_3^{(2)}) - C(Q_5^{(2)}) - 2C(Q_8^{(2)})$

21   $C(Q_{31}^{(2)}) \leftarrow \sum_{u \in V} \binom{N_u^{(2)}}{3} - C(Q_{20}^{(2)}) - C(Q_{23}^{(2)}) - 2C(Q_{12}^{(2)}) - C(Q_{14}^{(2)}) - 4C(Q_1^{(2)}) - 2C(Q_2^{(2)}) - C(Q_4^{(2)}) - C(Q_7^{(2)})$

22   $C(Q_{32}^{(2)}) \leftarrow \sum_{u \in V} \{\binom{|N_u^{(2)}|}{2} |N_u^{(1)}|\} - C(Q_{21}^{(2)}) - C(Q_{22}^{(2)}) - C(Q_{24}^{(2)}) - C(Q_{26}^{(2)}) - 2C(Q_{13}^{(2)}) - C(Q_{14}^{(2)}) - 2C(Q_{15}^{(2)}) - C(Q_{16}^{(2)}) - 2C(Q_{17}^{(2)})$

23       $-2C(Q_2^{(2)}) - 4C(Q_3^{(2)}) - 2C(Q_4^{(2)}) - 2C(Q_5^{(2)}) - 3C(Q_6^{(2)}) - C(Q_9^{(2)})$

24   $C(Q_{33}^{(2)}) \leftarrow \sum_{(u,v) \in E^{(2)}} \{(|N_u^{(2)}| - 1)(|N_v^{(2)}| - 1) - C_{(u,v)}(T_1^{(2)})\} - 4C(Q_{28}^{(2)}) - C(Q_{29}^{(2)}) - 2C(Q_{20}^{(2)}) - C(Q_{22}^{(2)}) - 6C(Q_{12}^{(2)}) - 4C(Q_{13}^{(2)})$

25       $-2C(Q_{14}^{(2)}) - C(Q_{15}^{(2)}) - C(Q_{16}^{(2)}) - 12C(Q_1^{(2)}) - 6C(Q_2^{(2)}) - 4C(Q_3^{(2)}) - 2C(Q_4^{(2)}) - C(Q_5^{(2)})$

26   $C(Q_{34}^{(2)}) \leftarrow \sum_{(u,v) \in E^{(2)}} \{|N_u^{(1)}|(|N_v^{(2)}| - 1) + (|N_u^{(2)}| - 1)|N_v^{(2)}| - \frac{1}{2}C_{(u,v)}(T_2^{(2)})\} - 2C(Q_{29}^{(2)}) - 2C(Q_{21}^{(2)}) - 2C(Q_{23}^{(2)}) - C(Q_{26}^{(2)})$

27       $-3C(Q_{14}^{(2)}) - 2C(Q_{16}^{(2)}) - 4C(Q_{17}^{(2)}) - 2C(Q_{18}^{(2)}) - 4C(Q_2^{(2)}) - 6C(Q_4^{(2)}) - 2C(Q_5^{(2)}) - 6C(Q_6^{(2)}) - 6C(Q_7^{(2)}) - 2C(Q_9^{(2)})$

28   $C(Q_{35}^{(2)}) \leftarrow \sum_{(u,v) \in E^{(1)}} (|N_u^{(2)}||N_v^{(2)}| - C_{(u,v)}(T_2^{(2)})) - C(Q_{29}^{(2)}) - 2C(Q_{30}^{(2)}) - C(Q_{22}^{(2)}) - 2C(Q_{25}^{(2)}) - 2C(Q_{13}^{(2)}) - C(Q_{14}^{(2)}) - 2C(Q_{15}^{(2)})$

29       $-2C(Q_{16}^{(2)}) - 3C(Q_{19}^{(2)}) - 2C(Q_2^{(2)}) - 4C(Q_3^{(2)}) - 2C(Q_4^{(2)}) - 3C(Q_5^{(2)}) - 4C(Q_8^{(2)})$

30   $C(Q_{36}^{(2)}) \leftarrow \sum_{(u,v) \in E^{(2)}} (|N_u^{(1)}||N_v^{(1)}| - C_{(u,v)}(T_3^{(2)})) - 2C(Q_{30}^{(2)}) - 2C(Q_{24}^{(2)}) - 3C(Q_{15}^{(2)}) - 2C(Q_{19}^{(2)}) - 4C(Q_3^{(2)}) - 3C(Q_5^{(2)}) - 4C(Q_8^{(2)})$

31       $-2C(Q_9^{(2)}) - 2C(Q_{10}^{(2)})$

---

**Table 6: Dataset statistics and exact counting time (sec.) for each algorithm. $|T^{(d)}|$ represents the number of instances of size-3 $(d, s)$-graphlets, and $|Q^{(d)}|$ represents the number of instances of size-4 $(d, s)$-graphlets. Additionally, the table includes the counting time for the conventional 4-size graphlet counting algorithms, PGD and ESCAPE, as well as the counting time for our method EDGE and the two baselines (EDGE-D2, EDGE-D).**

| Datasets | $\lvert T^{(2)}\rvert$ | $\lvert T^{(3)}\rvert$ | $\lvert Q^{(1)}\rvert$ | $\lvert Q^{(2)}\rvert$ | PGD | ESCAPE | EDGE-D2-(2,3) | EDGE-D-(2,3) | EDGE-(2,3) | EDGE-D2-(3,3) | EDGE-D-(3,3) | EDGE-(3,3) | EDGE-(2,4) |
|---|---|---|---|---|---|---|---|---|---|---|---|---|---|
| ca-DBLP | 4.89B | 769B | 629M | 3.95T | 0.381 | 0.297 | 82.3 | 17.5 | 4.31 | 19.0K | 4.64K | 913 | 62.5 |
| ca-Citeseer | 1.70B | 107B | 806M | 659B | 0.206 | 0.216 | 40.1 | 11.6 | 3.84 | 2.64K | 651 | 154 | 807 |
| ca-HepTh | 21.8M | 797M | 3.99M | 3.77B | 0.004 | 0.003 | 0.40 | 0.09 | 0.03 | 23.1 | 5.52 | 1.32 | 0.17 |
| web-Arabic | 755M | 12.6B | 779M | 205B | 0.427 | 1.25 | 23.2 | 7.58 | 2.72 | 308 | 69.0 | 18.0 | 607 |
| web-Indochina | 121M | 2.93B | 23.2M | 54.5B | 0.009 | 0.009 | 1.23 | 0.30 | 0.06 | 89.3 | 24.2 | 6.70 | 1.46 |
| soc-UCSC | 12.7B | 98.9B | 1.97B | 21.9T | 0.361 | 0.323 | 363 | 103 | 27.6 | 5.80K | 1.82K | 627 | 3.87K |
| soc-UC | 7.13B | 42.5B | 1.60B | 9.80T | 0.296 | 0.256 | 212 | 60.0 | 15.7 | 2.53K | 789 | 273 | 2.75K |
| soc-MB | 2.34B | 4.62B | 1.99B | 1.79T | 0.372 | 0.339 | 95.8 | 27.4 | 8.14 | 344 | 102 | 36.5 | 2.40K |
| tags-Ubuntu | 3.96B | 4.59B | 14.2B | 3.07T | 1.95 | 1.11 | 170 | 54.0 | 17.6 | 327 | 89.9 | 33.0 | 9.74K |
| tags-Math | 661M | 716M | 4.76B | 275B | 1.17 | 0.938 | 33.0 | 9.80 | 3.20 | 51.4 | 13.2 | 5.14 | 936 |
| road-CA | 45.0M | 189M | 14.0M | 301M | 0.312 | 0.285 | 7.76 | 7.66 | 7.30 | 19.0 | 17.5 | 15.5 | 7.88 |
| road-PA | 25.7M | 108.9M | 8.01M | 175M | 0.187 | 0.149 | 2.73 | 2.64 | 2.53 | 7.16 | 5.86 | 5.19 | 2.75 |
| road-TX | 30.7M | 128M | 9.52M | 202M | 0.238 | 0.188 | 4.24 | 4.20 | 3.96 | 10.4 | 9.33 | 8.2 | 4.31 |

