# OpenReview forum: "Beyond Neighbors: Distance-Generalized Graphlets for Enhanced Graph Characterization"
_ACM.org/TheWebConf/2025/Conference — WWW 2025 Poster_

### Official Review · Reviewer_HK7u · 2024-11-25

**Novelty:** 5
**Technical Quality:** 5

**Review:**

This paper proposes a novel concept, (d,s)-graphlet, to discover indirect connections between nodes. Based on specific (d,s)-graphlets, the new algorithm for exactly counting the occurrences is efficient. Both experimental and theoretical evidence is sufficient. I think that this work is of significance for graph characterization. I also noticed that the (d,s)-graphlet may lose efficacy due to high complexity when the distance d and graph size s increase. Despite this, the authors have made a great effort for specific implements and complexity analysis with the current technologies.

**Questions:**

(1)   The authors said, “Graphlets capture local structures within a graph, and real-world graphs can often be distinguished by their domain, or from random graphs, based on the occurrence patterns of the graphlets” (Line 46-48), and “In this work, we employ random graphs generated by the configuration model [45] as null graphs” (Line 255-257). However, the experiments are performed in 13 real-world databases. Do the authors distinguish real-world graphs from random graphs? I also wonder how the random graphs are configured. To my knowledge, a random graph is G(n,p), where n is the cardinality of the vertex set, and each node appears independently with constant probability p. Is the configuration model the same as G(n,p)?

(2)   The authors said, “Importantly, the sets 𝐸^(𝑑 ) are pairwise disjoint…” (Line 276-278). Can the authors explain it in more detail by using examples?

(3)   Why the authors categorized size-3 (𝑑, 𝑠)graphlets T^(𝑑 ) into two groups: non-deducible and deducible (𝑑, 𝑠)graphlet, while categorized size-4 (𝑑, 𝑠)graphlets into three groups: non-deducible, semi-deducible, and deducible (𝑑, 𝑠)graphlet? What is the definition of “deducible”?

(4)   The authors mentioned “clique” many times in Section 5 but did not give a definition of it. In Figure 3, it seems that the clique is under the distance d. It is better to give a formal definition.

(5)   The authors said, “There exists 6 (2,3)-graphlets, 13 (3,3)-graphlets and 36 (2,4)-graphlets” (Line 325-327). I notice that for specific d and s, the complexity exponentially increases according to Theorem 1 and 2. I wonder if a quantitative expression exists to calculate the number of (d,s) graphlets with respect to d and s. Are there any references that solved the problem? Is it an intractable problem?

**Reviewer Confidence:**

3: The reviewer is confident but not certain that the evaluation is correct

**Scope:**

4: The work is relevant to the Web and to the track, and is of broad interest to the community

---

### Official Review · Reviewer_NsPs · 2024-11-28

**Novelty:** 6
**Technical Quality:** 6

**Review:**

The paper introduces (d, s)-graphlets, a generalization of traditional graphlets that incorporates multi-hop relationships between nodes. The authors propose the EDGE algorithm, designed for efficiently counting (d, s)-graphlets with significantly improved performance compared to naive enumeration approaches. The method is empirically validated across various tasks, including graph clustering, characterization, and domain-specific analyses, demonstrating clear advantages over conventional graphlet-based methods.

The concept of distance-generalized graphlets addresses an important gap in graph analysis by accounting for indirect node relationships. The introduction of different deducibility levels for graphlets is particularly innovative, as it effectively reduces computational complexity.

Overall, the paper is well-written. However, it would benefit from additional illustrative examples to clarify key concepts, such as the differences between semi-deducible and non-deducible graphlets. Furthermore, the analysis focuses primarily on specific graphlet configurations, such as (2, 3), (3, 3), and (2, 4). Including discussions or experiments involving larger and more complex graphlets could strengthen the work and enhance its applicability.

**Questions:**

1. As mentioned above, the experiments focus on (2, 3), (3, 3), and (2, 4) graphlets. How does your method scale to configurations like (4,4) or even larger graphlets?

2. Have you considered benchmarking (d, s)-graphlets against other feature extraction techniques like graph neural networks? How would the performance and computational efficiency of your approach compare?

**Reviewer Confidence:**

2: The reviewer is willing to defend the evaluation, but it is likely that the reviewer did not understand parts of the paper

**Scope:**

4: The work is relevant to the Web and to the track, and is of broad interest to the community

---

### Official Review · Reviewer_mjev · 2024-12-03

**Novelty:** 3
**Technical Quality:** 4

**Review:**

Advantages:
1. New Insights in Real-World Data. Uncover the insights for counting (d, s)-graphlets with higher-order connections.
2. Comprehensive Experimental Evaluation: The authors conduct extensive experiments on 13 real-world datasets, evaluating the effectiveness of (d, s)-graphlets in graph characterization and clustering, as well as the speed and scalability of the EDGE algorithm.
3. Well-organized. The paper is well organized with clear thoughts.

Weaknesses:
1. Complexity of (d, s)-Graphlets. Computing (d, s)-graphlets, especially for larger values of d, is computationally more challenging than traditional graphlets due to the need to explore higher-order connections.
2. Limited Exploration of Larger Graphlets. While computing larger graphlets is time-consuming, the exploration of larger graphlets is relatively limited.
3. Lack of application. It’s recommended to make a further discussion on how to be adapted or extended to the algorithm in specific graph analysis.

**Questions:**

1. The paper mentions that the current focus is on size 3 and size 4. How difficult would it be to extend the proposed methods to other configurations, and what challenges might be encountered in terms of computational complexity and algorithm design? Or can we assume that the provided configurations is enough? How can we reach that assumption?
2. How could the proposed (d, s)-graphlets and the EDGE algorithm be adapted or extended to incorporate and utilize these attributes for more meaningful graph analysis?

**Reviewer Confidence:**

2: The reviewer is willing to defend the evaluation, but it is likely that the reviewer did not understand parts of the paper

**Scope:**

3: The work is somewhat relevant to the Web and to the track, and is of narrow interest to a sub-community